# Uncovering a 'sensitive window' of multisensory and motor neuroplasticity in the cerebrum and cerebellum of male and female starlings

Jasmien Orije[1]*, Emilie Cardon[1], Julie Hamaide[1], Elisabeth Jonckers[1], Veerle M Darras[2], Marleen Verhoye[1]*, Annemie Van der Linden[1]

[1]Bio-Imaging Lab, University of Antwerp, Antwerp, Belgium; [2]Laboratory of Comparative Endocrinology, Biology Department, Leuven, Belgium

**Abstract** Traditionally, research unraveling seasonal neuroplasticity in songbirds has focused on the male song control system and testosterone. We longitudinally monitored the song behavior and neuroplasticity in male and female starlings during multiple photoperiods using Diffusion Tensor and Fixel-Based techniques. These exploratory data-driven whole-brain methods resulted in a population-based tractogram confirming microstructural sexual dimorphisms in the song control system. Furthermore, male brains showed hemispheric asymmetries in the pallium, whereas females had higher interhemispheric connectivity, which could not be attributed to brain size differences. Only females with large brains sing but differ from males in their song behavior by showing involvement of the hippocampus. Both sexes experienced multisensory neuroplasticity in the song control, auditory and visual system, and cerebellum, mainly during the photosensitive period. This period with low gonadal hormone levels might represent a 'sensitive window' during which different sensory and motor systems in the cerebrum and cerebellum can be seasonally reshaped in both sexes.

*For correspondence:
Jasmien.Orije@uantwerpen.be
(JO);
marleen.verhoye@uantwerpen.be
(MV)

**Competing interests:** The authors declare that no competing interests exist.

## Introduction

Although various songbird species demonstrate different levels of female song, in one of the most studied song bird species, the zebra finch, female song is absent (*Odom et al., 2014*). Observations like these led to the discovery of one of the largest sexual dimorphisms in the brain of vertebrate species (*Bernard et al., 1993*; *Nottebohm and Arnold, 1976*). Such sexual dimorphisms in singing behavior and brain structure drove a bias in research of the song control system toward male songbirds (*Figure 1B*). Besides the large sexual dimorphism in song and song control nuclei, seasonal songbirds expose an additional peculiarity as they display naturally re-occurring seasonal cycles of singing and neuroplasticity of the song control system. However, the male focus proceeded in the research of seasonal neuroplasticity, since this process was shown to be largely driven by photoperiod-induced increases in testosterone, which results in a pre-optic area (POA)-mediated increase in motivation to sing and subsequent singing activity-induced neuroplasticity (*Alward et al., 2013*). The largest differences in song control nuclei volumes are found between breeding and non-breeding season (*Riters et al., 2002*). This led seasonal neuroplasticity studies to focus mainly on the breeding season or photostimulated phase, which they often compare to the non-breeding season or photorefractory and photosensitive phases. The main emphasis on the role of photoperiodic-induced increase in testosterone has led to many manipulation studies involving castration and/or testosterone implantation and its effect on song production and the song control system (e.g. *Hall and Macdougall-Shackleton, 2012*; *Orije et al., 2020*; *Stevenson and Ball, 2010*).

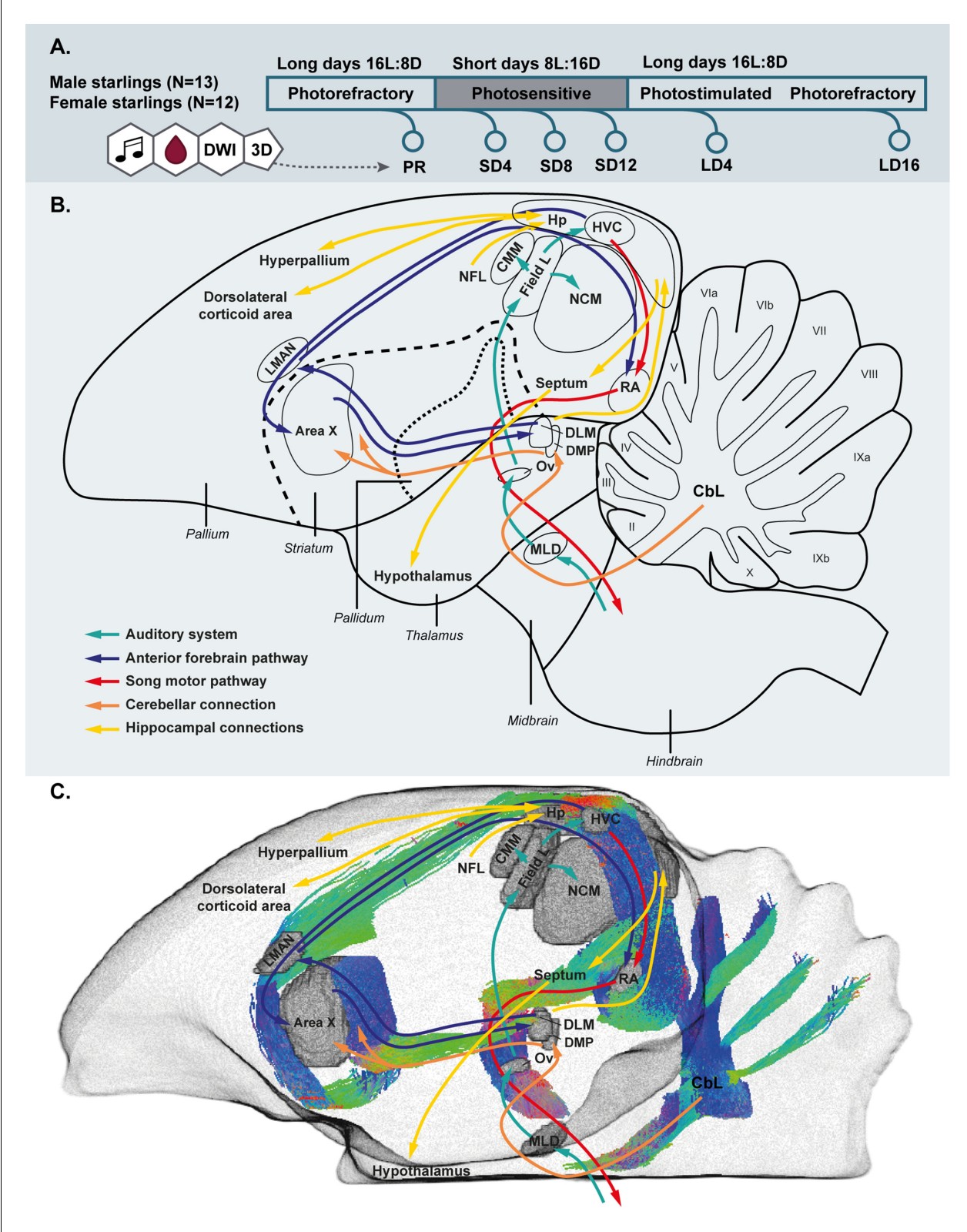

**Figure 1.** Simplified overview of the experimental setup. (**A**). Schematic overview of the song control and auditory system of the songbird brain and the cerebellar and hippocampal connections to the rest of the brain (**B**) and unilateral DWI-based 3D representation of the different nuclei and the interconnecting tracts as deduced from the tractogram (**C**). Male and female starlings were measured repeatedly as they went through different photoperiods. At each time point, their songs were recorded, blood samples were collected and T2-weighted 3D anatomical and diffusion weighted

*Figure 1 continued on next page*

*Figure 1 continued*

images (DWI) were acquired. The 3D anatomical images were used to extract whole brain volume (A). The song control system is subdivided in the anterior forebrain pathway (blue arrows) and the song motor pathway (red arrows). The auditory pathway is indicated by green arrows. The orange arrows indicate the connection of the lateral cerebellar nucleus (CbL) to the dorsal thalamic region further connecting to the song control system as suggested by *Person et al., 2008*; *Pidoux et al., 2018* (B,C). Nuclei in (C) are indicated in gray, the tractogram is color-coded according to the standard red-green-blue code (red = left right orientation (L-R), blue = dorso ventral (D-V) and green = rostro caudal (R-C)).

The online version of this article includes the following video for figure 1:

**Figure 1—video 1.** Movie of the unilateral 3D representation of the different nuclei and the interconnecting tracts rotating along the vertical axis.

https://elifesciences.org/articles/66777#fig1video1

Prior studies have also pointed out that testosterone implants only have an effect when the birds have experienced a photosensitive phase with short day length exposure (*Bernard and Ball, 1997*; *Rouse et al., 2015*). Although little research is devoted to brain neuroplasticity during this state (*Larson et al., 2019*; *Riters et al., 2002*), multiple studies have shown that during the photosensitive period, testes size (*Cornez et al., 2017*) and GnRH expression in the POA slowly start to increase (*Bentley et al., 2013*; *Hurley et al., 2008*; *Stevenson et al., 2012*), while some song control nuclei slowly start to grow (*Riters et al., 2002*). The photosensitive period might be seen as a sensitive window, defined as a period in which experiences lead to long-lasting changes (that remain in the subsequent photostimulated phase or breading season) in the brain and song behavior (*Knudsen, 2004*). The photosensitive period plays a crucial role in creating permissive circumstances for enhanced neuroplasticity resulting in enlarged song control nuclei as observed during the photostimulated stage.

The studies on sexual dimorphisms and seasonal neuroplasticity in songbirds have been preoccupied with vocal control areas as the neural substrate for the behavioral sexual dimorphism in song performance. However, in vivo Magnetic Resonance Imaging (MRI) allows us to visualize the entire brain and study the involvement of other structures like the hippocampus and cerebellum, which recently have been shown to be involved in song preference and song processing, respectively (*Bailey et al., 2009*; *Person et al., 2008*; *Pidoux et al., 2018*; *Striedter, 2016*). Prior MRI studies from our team focusing on the entire songbird brain have demonstrated seasonal neuroplasticity and functional changes in both auditory and olfactory systems of starlings (*De Groof et al., 2017*; *De Groof et al., 2010*; *De Groof et al., 2013*). This outcome hints toward a seasonal re-occurring cycle of multisensory neuroplasticity and raises the question whether the photosensitive stage perhaps represents a renewed 'sensitive window' for heightened neuroplasticity and rewiring of various sensory and sensorimotor systems.

The current study aims to fill the gaps highlighted above and answers the following questions with an extensive data-driven exploratory brain-wide structural MRI study of both male and female starlings throughout different seasons: (1) Is the sexual dimorphism limited to the song control system in starlings, as in zebra finches where only the male sings (*Hamaide et al., 2017*) or does this dimorphism extend to other sensory systems? Can it be attributed to a difference in total brain size between the sexes, similar to what has been observed in mammals? (2) Do female and male starlings display parallel or different seasonal structural neuroplasticity? Is this neuroplasticity limited to the song control system or does it extend to other neural circuits? When do these (neuroplasticity) changes take place, already during the photosensitive period when gonadal hormone levels are still very low? (3) How does seasonal neuroplasticity correlate – at the group and individual level – with gonadal hormone levels and with song-production?

Since MRI is an in vivo method, we were able to measure the same subjects repeatedly and monitor the structural neuroplasticity longitudinally in 13 male and 12 female starlings as they undergo different subsequent photoperiodic phases: at baseline photorefractory state (PR), after 4, 8, and 12 weeks of short days during the photosensitive state (SD4, SD8, SD12), after 4 weeks of photostimulation (LD4) and after 16 weeks on long days when the birds are photorefractory again (LD16) (*Figure 1A*). At each time point we: collected blood samples; measured body weight; recorded song; and acquired structural MRI data including 3D anatomical scans and diffusion weighted images (DWI). Using voxel-based diffusion tensor imaging (DTI) and fixel-based analysis (*De Groof et al., 2006*; *Raffelt et al., 2017*), we could visualize different sensory and sensorimotor circuits and their

mutual connections (*Figure 1C*). We established spatio-temporal statistical maps revealing micro-structural changes in gray and white matter, specifically in fiber connections throughout the different seasons in the entire brain in both sexes. To learn specifically about the contribution of singing activity-induced neuroplasticity in the seasonal neuroplasticity, a voxel-based multiple regression between seasonal song rate and corresponding DTI parameter maps was performed to identify the neuronal correlates of song behavior in both sexes (*Hamaide et al., 2020*; *Sagi et al., 2012*).

## Results

### Population-based tractogram of the starling brain using fixel-based analysis

In the current study, we analyzed the DWI scans in two distinct ways: (1) using the common approach of diffusion tensor derived metrics such as fractional anisotropy (FA) and; (2) using a novel method of fiber orientation distribution (FOD)-derived fixel-based analysis. Both techniques infer the micro-structural information based on the diffusion of water molecules, but they are conceptually different (*Table 1*). Common DTI analysis extracts for each voxel several diffusion parameters, which are sensitive to various microstructural changes in both gray and white matter specified in *Table 1*. Fixel-based analysis on the other hand explores both microscopic changes in apparent fiber density (FD) or macroscopic changes in fiber-bundle cross-section (log FC) (*Table 1*). Positive fiber-bundle cross-section values indicate expansion, whereas negative values reflect shrinkage of a fiber bundle relative to the template (*Raffelt et al., 2017*).

A population-based template created for the fixel-based analysis can be used as a study-based atlas in which many of the avian anatomical structures can be identified (*Figure 2*). We recognize many of the white matter structures such as the different lamina, occipito-mesencephalic tract (OM) and optic tract (TrO) among others. Interestingly, many of the nuclei within the song control system (i.e. HVC, robust nucleus of the arcopallium (RA), lateral magnocellular nucleus of the anterior nido-pallium (LMAN), and Area X), auditory system (i.e. intercollicular nucleus complex, nucleus ovoidalis) and visual system (i.e. entopallium, nucleus rotundus) are identified by the empty spaces between tracts. The applied fixel-based approach is inherently sensitive to changes in white matter and cannot report on the microstructure within gray matter like brain nuclei; but rather sheds light on the fiber tracts surrounding and interconnecting them. As such, it provides an excellent tool to investigate neuroplasticity of different brain networks, and in the case of a nodular song control system

**Table 1.** Overview of the parameters studied by common DTI analysis and novel fixel-based analysis, including a comprehensive, but not exhaustive, list of some of the microstructural changes to which they are sensitive.

| | Metric | Level | Para-meter | Measures | Sensitive to changes in: |
|---|---|---|---|---|---|
| DTI analysis | Diffusion tensor | Voxel | FA | Directionality of water diffusion | Axon number and density, axon diameter, myelination, fiber organization |
| | | | MD | Average diffusion across all directions | Cell size, cell spacing, cell density, dendrite branching, extracellular space |
| | | | AD | Diffusion along the fiber direction | Axon number and density |
| | | | RD | Diffusion perpendicular to the fiber direction | Axon diameter and myelination |
| Fixel-based analysis | Fiber orientation distribution | Fixel | FD | Microscopic density within fiber population | Axon number and density, axon diameter |
| | | | FC | Macroscopic change in cross-sectional area perpendicular to fiber bundle | Fiber bundle size due to changes in extra-axonal space and/or myelination, number of axons |

FA, fractional anisotropy; MD, mean diffusivity; AD, axial diffusivity; RD, radial diffusivity; FD, fiber density; FC, fiber-bundle cross-section. Fixel-based analysis can distinguish between different fiber bundles within a voxel and estimate fiber density for each fiber bundle separately (FD1 and FD2). For a complete description and interpretation of these measures we refer to the following reviews of DTI analysis (*Beaulieu, 2002*; *Beaulieu, 2013*; *Song et al., 2002*; *Zatorre et al., 2012*) and fixel based analysis (*Genc et al., 2020*; *Raffelt et al., 2017*).

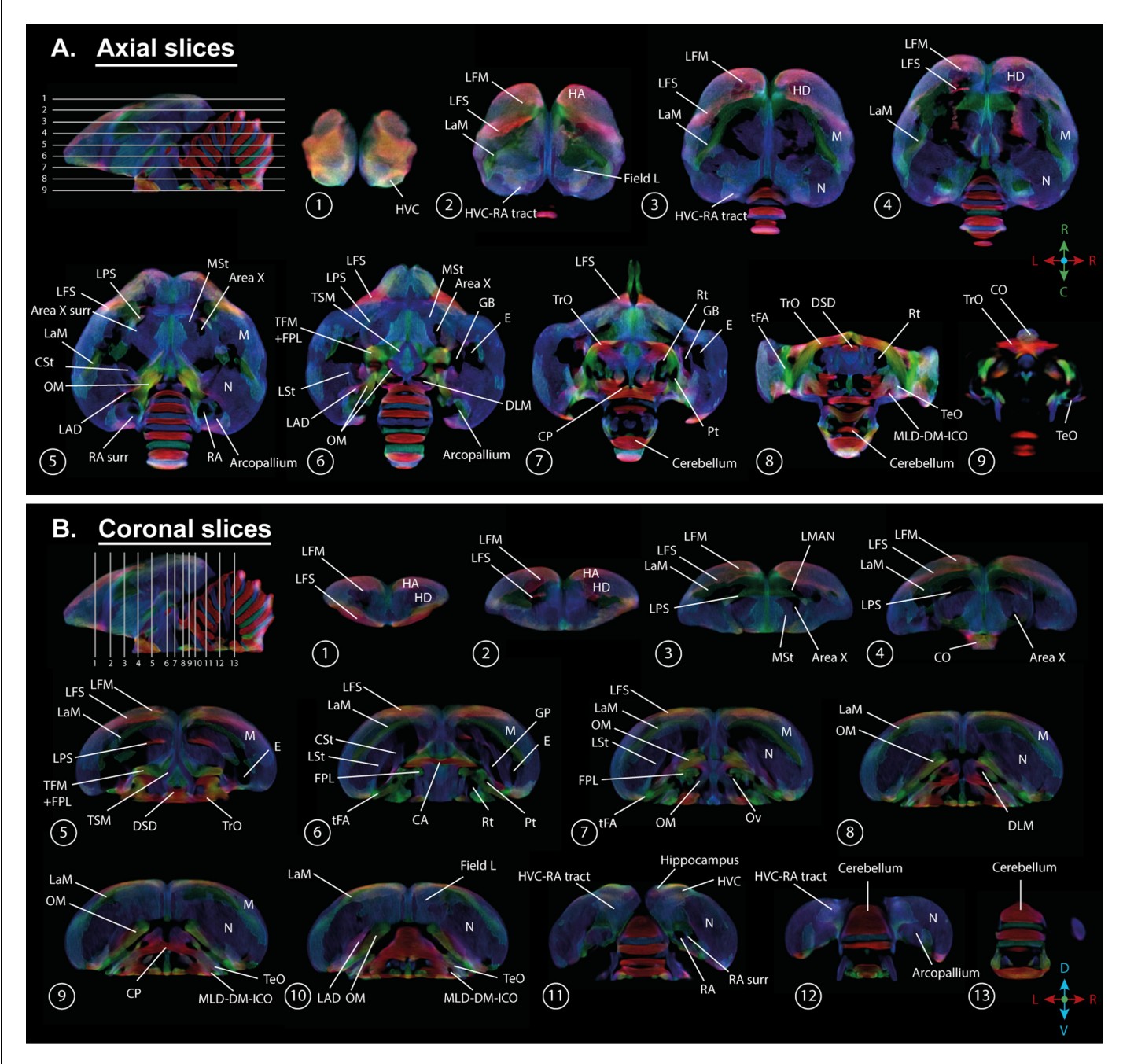

**Figure 2.** Overview of the population-based tractogram of male and female starlings over the seasons created for the fixel-based analysis with indications of different lamina, interconnecting tracts, nuclei, and brain regions displayed on axial (A) and coronal slices (B) throughout the brain. The different intervals of the coronal and axial sectioning are indicated on the saggittal inset in the left corner of each panel. The coronal slices do not follow a strict interval to visualise small nuclei such as DLM and Ov. The crosshair indicates the color-coding of the tractogram following the standard red-green-blue code (red = left right orientation (L-R), blue = dorso ventral (D-V) and green = rostro caudal (R-C)).

The online version of this article includes the following video for figure 2:

**Figure 2—video 1.** Movie of the population-based tractogram of male and female starlings scanning through the coronal slices with the different lamina, interconnecting tracts, nuclei, and brain regions indicated.

https://elifesciences.org/articles/66777#fig2video1

focusing on changes in the fibers surrounding the song control nuclei, referred to as HVC surr, RA surr, and Area X surr.

## Data-driven analysis of sexual dimorphisms in brain structure

We established the general structural differences between male and female starling brains by applying a full-factorial design voxel-based analysis on our DTI data (*Figures 3* and *4*, *Table 2*). This includes data from all time points with time as a within-subject factor that needs to be averaged over (*McFarquhar, 2019*). This analysis confirmed the well-established sexual dimorphism of vocal control areas in songbirds (*Bernard et al., 1993*; *Hamaide et al., 2017*; *Nottebohm and Arnold, 1976*), where male starlings have higher fractional anisotropy in several parts of the song control system (the surroundings of HVC, Area X and RA) and auditory system (Field L and caudomedial nidopallium [NCM]) compared to female starlings. Interestingly, even though most differences occur bilaterally, these structures are visible as sub-peaks within a single significant cluster. This cluster was asymmetric toward the left hemisphere and comprised large parts of the mesopallium and nidopallium, but not the lamina mesopallialis (LaM) dividing these regions.

Importantly, females generally have higher fractional anisotropy in tracts interconnecting both hemispheres (commisura anterior and posterior) and in bilateral tracts interconnecting nuclei within a hemisphere including the TrO, OM, septopallio-mesencephalic tract (TSM), and fronto-arcopallial tract (tFA). In addition, females have higher fractional anisotropy within the cerebellum and caudal part of the lateral striatum (CSt) compared to males. These findings are further confirmed by fixel-based analysis; as many of the sexual dimorphisms in fractional anisotropy match with the fiber density statistical analysis (*Figure 4A,B*). Certain subregions of the cerebellum, the molecular layer of cerebellar lobules II, III, IV have higher fiber density in females, whereas the cerebellar lobule VIII has higher fiber density in males. This diversifies the finding of increased fractional anisotropy in the cerebellum of female starlings. The molecular layer of the cerebellum consists of parallel fibers emitted by the granule cells of the cerebellar cortex (*D'Angelo, 2018*). Fixel-based analysis is able to detect

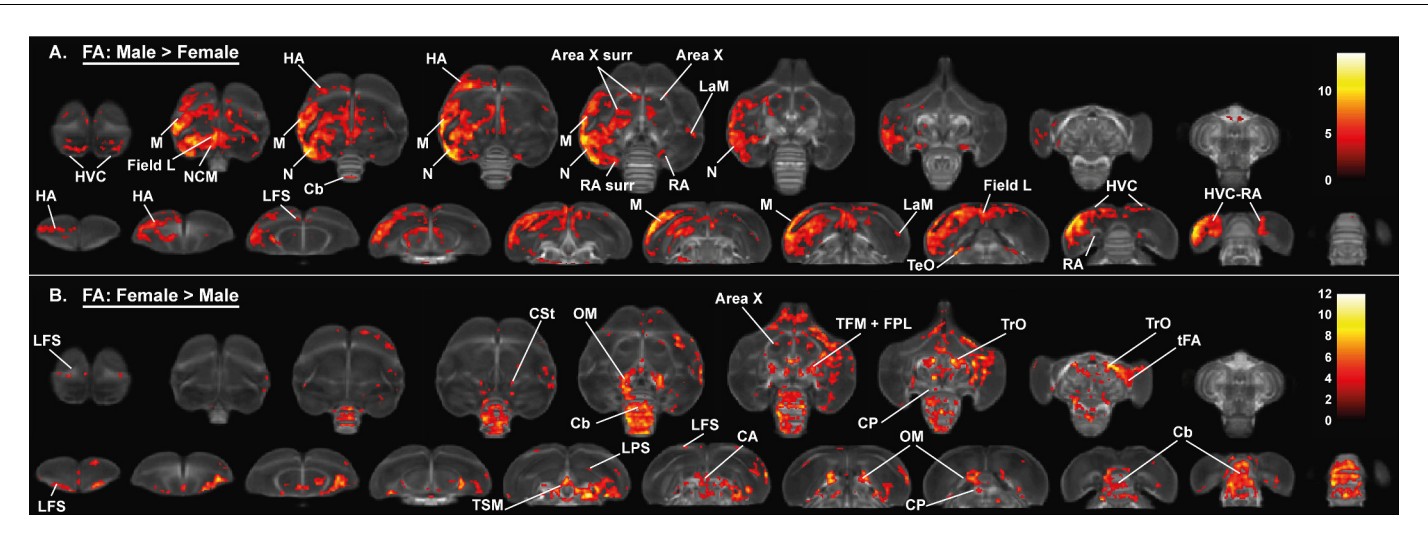

**Figure 3.** Overview of the general structural sexual dimorphism within the starling brain attributed to differences in fractional anisotropy (FA). For each sex comparison, the statistical parametric maps are displayed on axial and coronal sections throughout the brain (upper and lower row, respectively). The results are displayed with $p_{uncorr}$ <0.001 and $k_E \geq 10$ voxels, and overlaid on the population fractional anisotropy map. The T-values are color-coded according to the scale on the right. In males, fractional anisotropy was higher in a large cluster lateralized to the left hemisphere and covering parts of the nidopallium and mesopallium, including several regions of the auditory system and surrounding the song control system. Females, on the other hand, have higher fractional anisotropy at the level of the cerebellum and several tracts such as OM, TSM, and TrO.

The online version of this article includes the following source data and figure supplement(s) for figure 3:

**Figure supplement 1.** Overview of the fractional anisotropy (FA) changes over time extracted from the relevant ROI-based clusters with significant sex differences.

**Figure supplement 1—source data 1.** This file contains the source data used to make the graphs presented in *Figure 3—figure supplement 1*.

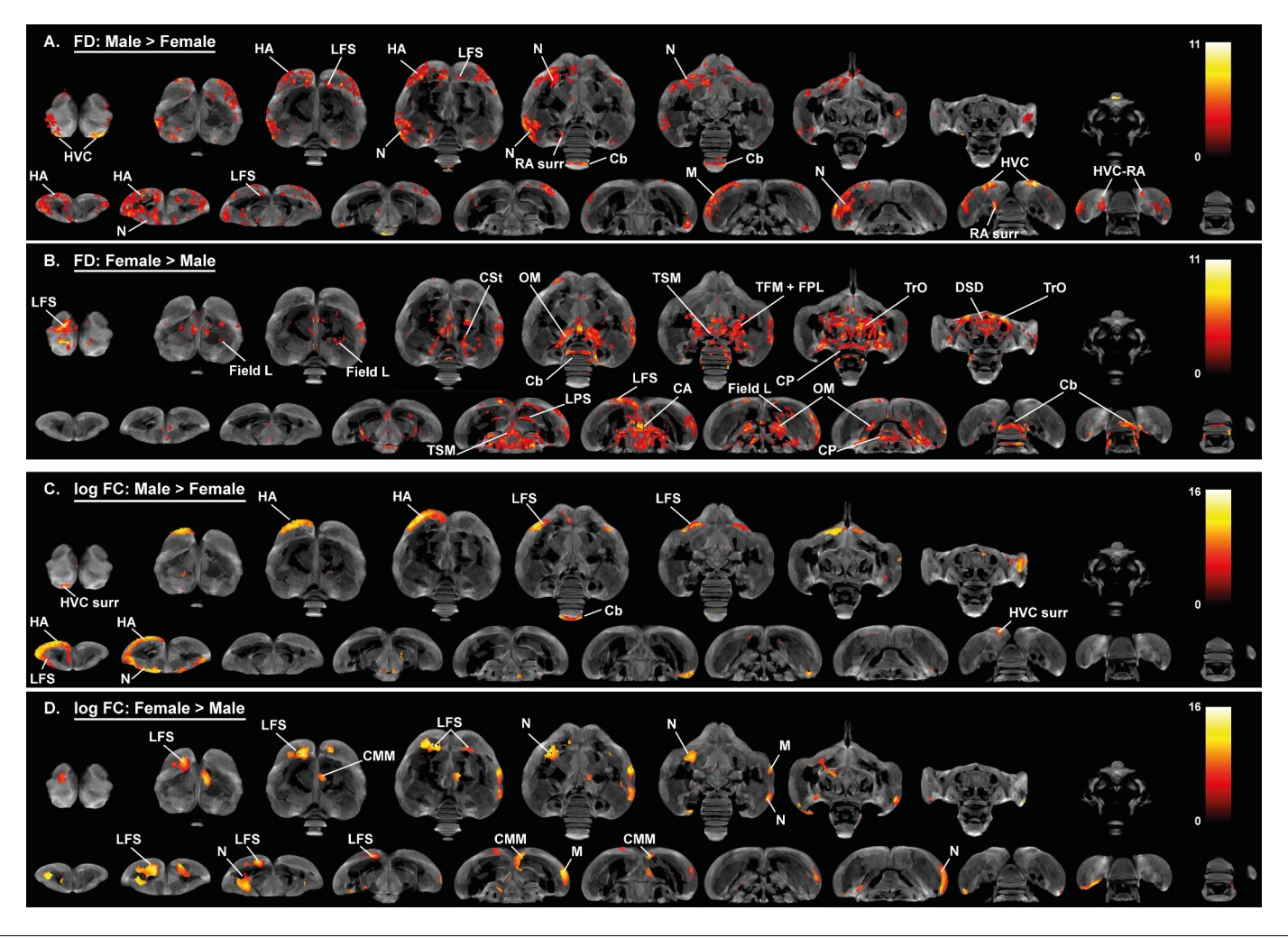

**Figure 4.** Overview of the general structural sexual dimorphism within the starling brain attributed to microscopic differences in fiber density (FD) (**A, B**) and macroscopic differences in fiber-bundle cross-section (log FC) (**C, D**). For each sex comparison, the statistical parametric maps are displayed on axial and coronal sections throughout the brain (upper and lower row respectively). The results are displayed with $p_{uncorr}$ <0.001, and overlaid on the population based tractogram. Only significant tracks are displayed in a color representing the T-value.

The online version of this article includes the following source data and figure supplement(s) for figure 4:

**Source data 1.** This file contains the source data used to make the graphs presented in *Figure 4—figure supplement 2* and *3*.

**Figure supplement 1.** Overview of the general structural sexual dimorphism within the starling brain attributed to microscopic differences in fiber density (FD) (**A, B**) and macroscopic differences in fiber-bundle cross-section (log FC) (**C, D**) using fiber orientation color coding.

**Figure supplement 2.** Overview of the fiber density (FD) changes over time extracted from the relevant ROI-based clusters with significant sex differences.

**Figure supplement 3.** Overview of the fiber-bundle cross-section (log FC) changes over time extracted from the relevant ROI-based clusters with significant sex differences.

significant changes in this highly organized fiber structure, which is depicted as the red parallel fibers running left to right in *Figure 4—figure supplement 1B*. In addition to these microscopic changes, fiber-bundle cross-section statistical analysis revealed some macroscopic differences between male and female starlings (*Figure 4C,D*). Male starlings generally have a larger volume in the rostral part of the left hemisphere at the level of the hyperpallium apicale, which further supports the asymmetry toward the left hemisphere. Furthermore, the left HVC surroundings and cerebellar lobule VIII are larger in males compared to females. However, certain regions within the superior frontal lamina (LFS), nidopallium and mesopallium are larger in females, compared to males. Extracted values from

**Table 2.** Clusters displaying a sex difference in fractional anisotropy (FA).

Table summarizing the significance at cluster level and peak level. Some clusters are large and cover multiple regions, indicated by the single statistic at cluster level. For each sub-region within this large cluster the peak significance (pFWE) and T-value (T) are reported. p-Values are FWE corrected. $K_E$ indicates the number of continuous voxels within a cluster.

| Main sex difference for FA | Cluster | Hemisphere | Cluster | | Peak | |
| --- | --- | --- | --- | --- | --- | --- |
| | | | $p_{FWE}$ | $k_E$ | $p_{FWE}$ | $T$ |
| Male > female | Nidopallium/mesopalium | Left | <0.0001 | 36936 | <0.0001 | 14.25 |
| | HVC surr | Left | | | <0.0001 | 8.35 |
| | | Right | | | <0.0001 | 7.53 |
| | Field L dorsal | Left | | | <0.0001 | 10.00 |
| | | Right | | | <0.0001 | 6.93 |
| | NCM | Left | | | <0.0001 | 7.72 |
| | | Right | | | <0.0001 | 6.72 |
| | Area X caudal surr | Left | | | 0.001 | 5.69 |
| | | Right | | | 0.001 | 6.07 |
| | RA surr | Left | | | <0.0001 | 7.86 |
| | HVC-RA tract | Left | | | <0.0001 | 6.68 |
| | RA surr | Right | <0.0001 | 441 | 0.001 | 6.08 |
| | HVC-RA tract | Right | | | <0.0001 | 6.42 |
| | TeO (superior) | Left | <0.0001 | 189 | <0.0001 | 9.44 |
| | | Right | <0.0001 | 125 | <0.0001 | 7.47 |
| | LaM | Right | <0.0001 | 165 | <0.0001 | 6.97 |
| | Cerebellum | Lobule VII | 0.011 | 101 | <0.0001 | 6.54 |
| Female > male | TrO | Right | <0.0001 | 24846 | <0.0001 | 12.08 |
| | | Left | | | <0.0001 | 8.84 |
| | TSM | Left | | | <0.0001 | 9.62 |
| | | Right | | | <0.0001 | 9.74 |
| | Commisura anterior | Center | | | 0.222 | 4.85 |
| | Commisura posterior | Center | | | 0.066 | 5.19 |
| | OM | Left | | | <0.0001 | 8.77 |
| | | Right | | | <0.0001 | 7.78 |
| | CSt | Left | | | <0.0001 | 8.50 |
| | | Right | | | <0.0001 | 7.64 |
| | Cerebellum | Lobule VIII | | | <0.0001 | 9.12 |
| | | Lobule VII | | | <0.0001 | 9.68 |
| | | Lobule VI | | | <0.0001 | 7.77 |
| | | Lobule V | | | <0.0001 | 7.12 |
| | | Lobule IV | | | <0.0001 | 6.58 |
| | tFA | Right | | | <0.0001 | 10.50 |
| | | Left | 0.006 | 113 | <0.0001 | 8.64 |
| | Area X | Left | 0.139 | 57 | <0.0001 | 6.34 |
| | | Right | <0.0001 | 254 | <0.0001 | 6.91 |

the identified clusters are plotted to demonstrate the extent of the sex difference in the respective diffusion parameters (*Figure 3—figure supplement 1*, *Figure 4—figure supplements 2* and *3*).

Alongside the well-established difference in the song control system, the applied data-driven approach sheds new light on sexual dimorphisms in the rest of the starling brain, including a

microstructural hemispheric asymmetry toward the left hemisphere in males and a more structurally organized cerebellum as well as several white matter tracts in females.

## Role of brain size in the general sexual dimorphism

The hypothesis of neuronal interconnectivity is used to explain some of the sexual dimorphisms in brain structure of mammals and postulates that larger brains have relatively stronger intrahemispheric connections, resulting in stronger lateralization in larger brains, whereas smaller brains benefit more from higher interhemispheric connectivity (*Hänggi et al., 2014*). To ensure that the general sexual dimorphisms are attributed to a genuine sex difference and not to a difference in brain size, we examined: (1) whether there is a difference in brain size between males and females and; (2) if this difference in brain size can explain sexual dimorphisms like the higher interhemispheric interconnectivity found in females. Even though there was a small difference in brain size between male and female starlings (respectively, Mean male = 1815, SE = 8 mm$^3$ and Mean female = 1770, SE = 13 mm$^3$), it was not significant (F(1, 23)=1.78, p=0.195). Next, we artificially divided each sex in two groups based on their brain size using a median split similar to the analysis in *Kurth et al., 2018*.

These artificially generated groups were used for a voxel-based analysis with brain size as a fixed factor to assess the effect of brain size on the fractional anisotropy, fiber density and fiber-bundle cross-section analysis (*Figures 5* and *6*, *Figure 6—figure supplement 1*, *Table 3*). This analysis could not explain the hemispheric asymmetry in males nor the increased interhemispheric connections in females. Only a few sex differences matched with the voxel-based analysis for brain size. Large brain male and female starlings had higher fractional anisotropy values in specific sections of the song control system including the caudal surroundings of the right Area X, surroundings of RA and HVC (*Figure 5A*). The brain size difference in fractional anisotropy in right Area X surroundings and cerebellum was matched with a difference in fiber density (*Figure 6A*), whereas the higher fractional anisotropy in the surroundings of HVC of large brain starlings, is complemented by a higher fiber-bundle cross-section in the surroundings of HVC (*Figure 6C*).

For the neuronal interconnectivity hypothesis to hold up in starlings, we expected that small brain starlings exhibit higher fractional anisotropy in several interconnecting tracts. However, only a small

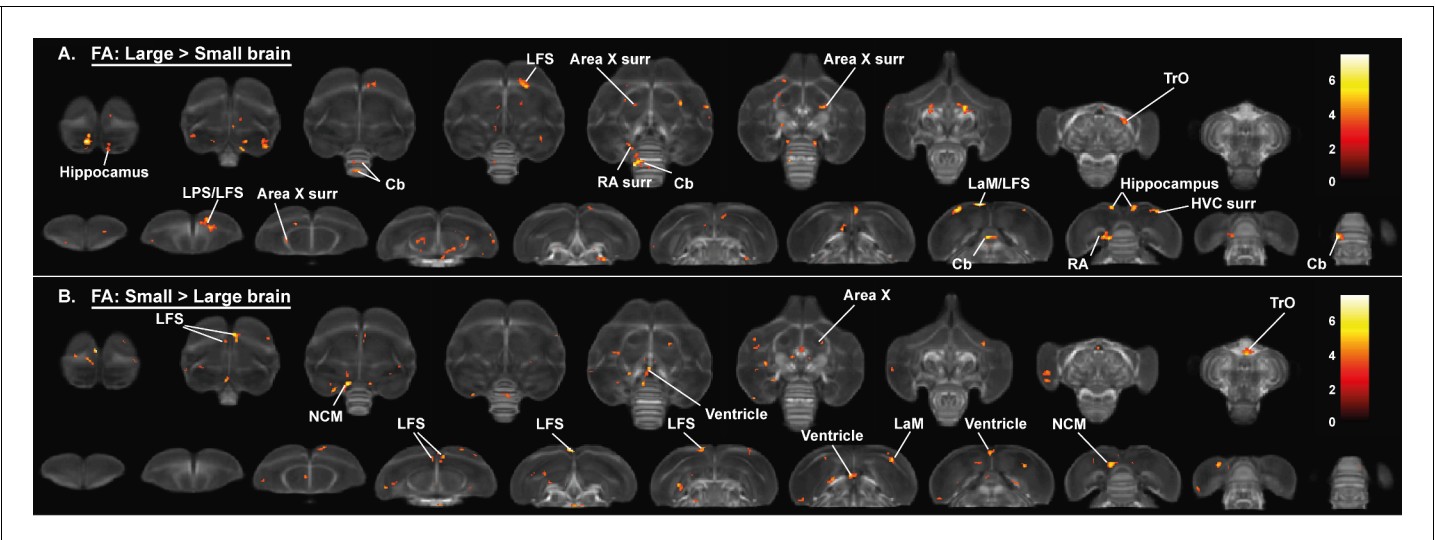

**Figure 5.** Overview of general structural difference between large and small brain starling in fractional anisotropy (FA). The statistical parametric maps are displayed on axial and coronal sections throughout the brain (upper and lower row, respectively). The results are displayed with p$_{uncorr}$ <0.001 and k$_E$ ≥ 10 voxels, and overlaid on the population fractional anisotropy map. The T-values are color-coded according to the scale on the right.

The online version of this article includes the following source data and figure supplement(s) for figure 5:

**Figure supplement 1.** Overview of the fractional anisotropy (FA) changes over time extracted from the relevant ROI-based clusters with significant differences in brain size.

**Figure supplement 1—source data 1.** This file contains the source data used to make the graphs presented in *Figure 5—figure supplement 1*.

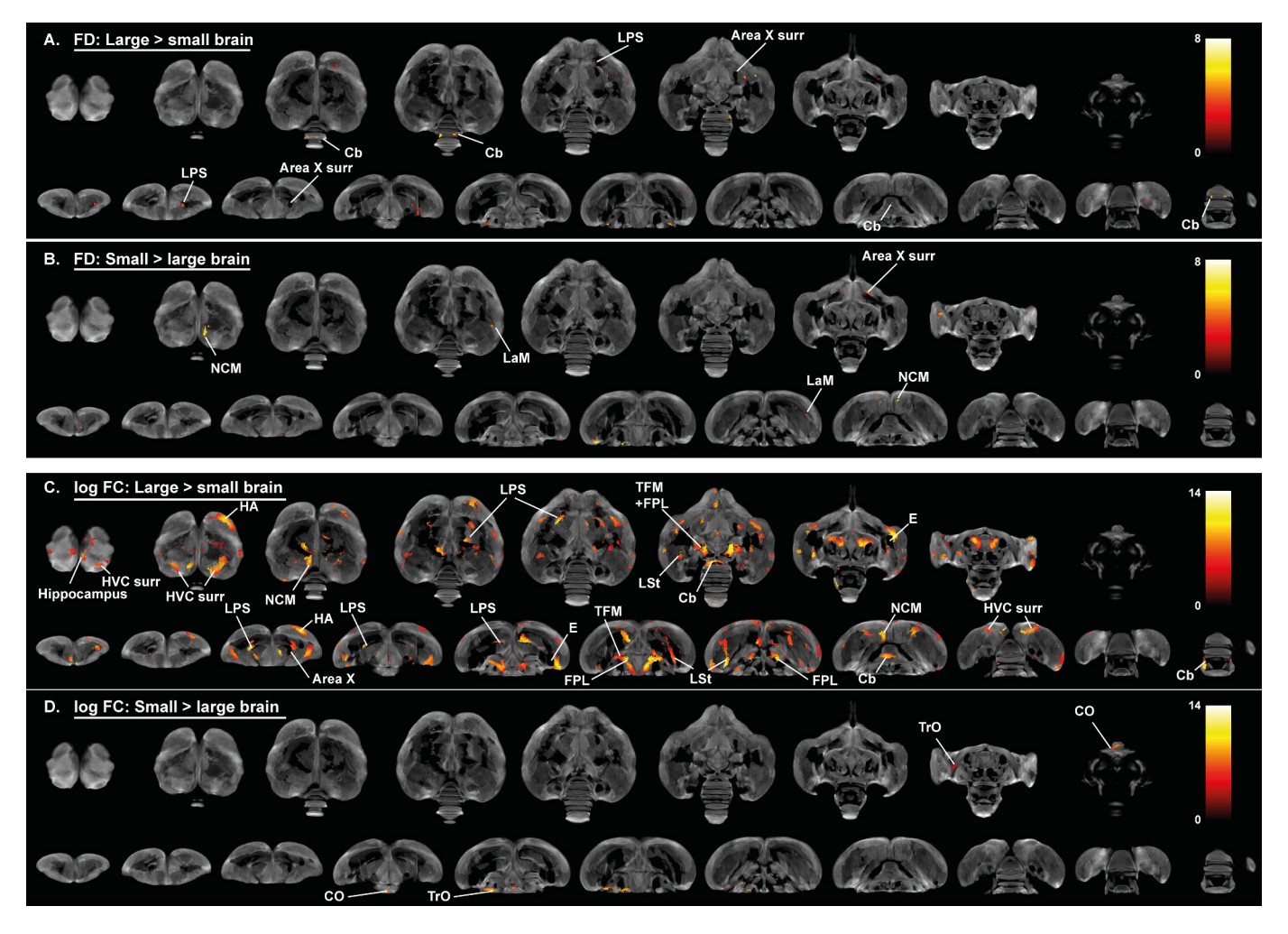

**Figure 6.** Overview of general structural difference between large and small brain starling in microscopic differences in fiber density (FD) (**A, B**) and macroscopic differences in fiber-bundle cross-section (log FC) (**C, D**). The statistical parametric maps are displayed on axial and coronal sections throughout the brain (upper and lower row respectively). The results are displayed with p$_{uncorr}$ <0.001, and overlaid on the population-based tractogram. Only significant tracks are displayed and colored according to their significance by T-value.

The online version of this article includes the following source data and figure supplement(s) for figure 6:

**Source data 1.** This file contains the source data used to make the graphs presented in *Figure 6—figure supplement 2* and *3*.

**Figure supplement 1.** Overview of general structural difference between large and small brain starling in microscopic differences in fiber density (FD) and macroscopic differences in fiber-bundle cross-section (log FC) using fiber orientation color coding.

**Figure supplement 2.** Overview of the fiber density (FD) changes over time extracted from the relevant ROI-based clusters with significant differences in brain size.

**Figure supplement 3.** Overview of the fiber-bundle cross-section (log FC) changes over time extracted from the relevant ROI-based clusters with significant differences in brain size.

part of the TrO had higher fractional anisotropy values in small brain starlings (*Figure 5A*), which closely matches the brain size difference of fiber-bundle cross-section in TrO (*Figure 6D*). In left NCM and a right LaM, both male and small brain starlings exhibit higher fractional anisotropy and fiber density values (*Figures 5A* and *6B*), which might be partially explained by the finding of higher fiber-bundle cross-section values in the NCM of large compared to small brain starlings (*Figure 6C*). Furthermore, some regions such as the hippocampus in large brain starlings and ventricle in small brain birds were not detected in the voxel-based sex difference analysis and are solely attributed to a difference in brain size. In the fiber-bundle cross-section analysis for brain size differences, more regions unrelated to a sex difference showed a significant larger fiber-bundle cross-section value in

**Table 3.** Clusters displaying a difference in brain size versus a difference in sex in fractional anisotropy (FA).

Table summarizing the significance at cluster level and peak level. Some clusters are large and cover multiple regions, indicated by the single statistic at cluster level. For each sub-region within this large cluster the peak significance (pFWE) and T-value (T) are reported. p-Values are FWE corrected. $K_E$ indicates the number of continuous voxels within a cluster. The final column reports if a region with a difference in brain size is analogue to other relevant differences such as the general sexual dimorphism in fractional anisotropy (M > F or F > M) or the correlation between song rate and fractional anisotropy in females (↑Singing F).

| | Cluster | Hemisphere | Cluster | | Peak | | Relevant difference |
|---|---|---|---|---|---|---|---|
| | | | $p_{FWE}$ | $k_E$ | $p_{FWE}$ | T | |
| | HVC surr | Left | <0.0001 | 227 | <0.0001 | 6.43 | M > F |
| | Hippocampus | Left | | | <0.0001 | 5.63 | ↑singing F |
| | LaM/LFS | Left | <0.0001 | 155 | <0.0001 | 7.51 | |
| | HVC surr | Right | | | <0.0001 | 6.61 | M > F |
| | Hippocampus | Right | <0.0001 | 417 | <0.0001 | 6.94 | |
| | Area X caudal surr | Right | <0.0001 | 190 | 0.002 | 5.99 | M > F<br>↑Singing F |
| | LPS | Left | 0.004 | 126 | 0.006 | 5.78 | |
| | Lps/LFS | Right | <0.0001 | 337 | 0.042 | 5.29 | |
| | tFA | Right | 0.925 | 20 | 0.029 | 5.39 | F > M |
| | TrO | Right | 0.001 | 160 | <0.0001 | 7.29 | F > M |
| | RA surr | Left | | | 0.193 | 4.88 | M > F |
| | Cerebellum | Lobule VI | <0.0001 | 451 | <0.0001 | 7.50 | |
| | Cerebellum | Lobule VI | 0.259 | 48 | 0.010 | 5.64 | |
| | Cerebellum | Lobule IV | 0.114 | 62 | 0.003 | 5.94 | |
| Large > small brain | Cerebellum | Lobule VII | 0.101 | 64 | 0.041 | 5.30 | |
| Small > large brain | NCM | Left | 0.006 | 116 | <0.0001 | 6.39 | M > F |
| | LaM | Right | 0.014 | 100 | 0.025 | 5.43 | M > F |
| | Ventricle rostral | Left | 0.217 | 51 | <0.0001 | 7.21 | |
| | | Right | <0.0001 | 246 | <0.0001 | 6.83 | |
| | Ventricle caudal | Center | 0.002 | 138 | 0.003 | 5.90 | |
| | | Center | 0.004 | 122 | 0.214 | 4.84 | |
| | TrO | Center | 0.011 | 105 | 0.232 | 4.82 | F > M |

large brains compared to smaller brains, including parts of the pallial-subpallial lamina, lateral fore-brain bundle, medial and lateral surroundings of the entopallium.

Interestingly, many of the fiber tracts that had a pronounced sex difference, such as the anterior and posterior commissure, TSM and OM, were not found in the brain size statistical maps. Furthermore, the microstructural hemispheric asymmetry covering the nidopallium and mesopallium detected in the voxel-based sex difference analysis can also not be attributed to a mere difference in brain size. Importantly, further statistical analysis is required to ensure that the brain size effect is not confounded by other factors such as song behavior differences.

## Sexual dimorphism in seasonal neuroplasticity

After establishing the general structural sexual dimorphisms, we investigated whether seasonal neuroplasticity occurs differently between sexes. Therefore, we performed a voxel-based flexible factorial analysis looking at the general fractional anisotropy changes over time and the interactions between time and sex. These results are summarized in *Figure 7* and *Table 4*. We extracted the mean fractional anisotropy values from the significant clusters and plotted them to determine how fractional anisotropy changes over time (*Figure 8*). Fractional anisotropy changes can be caused by a multitude of underlying microstructural changes, however fractional anisotropy is most sensitive to changes in axon number and density, axon diameter and myelination (see *Table 1*). Additionally, we extracted the mean of the other diffusion parameters (mean, axial and radial diffusion) and fixel-

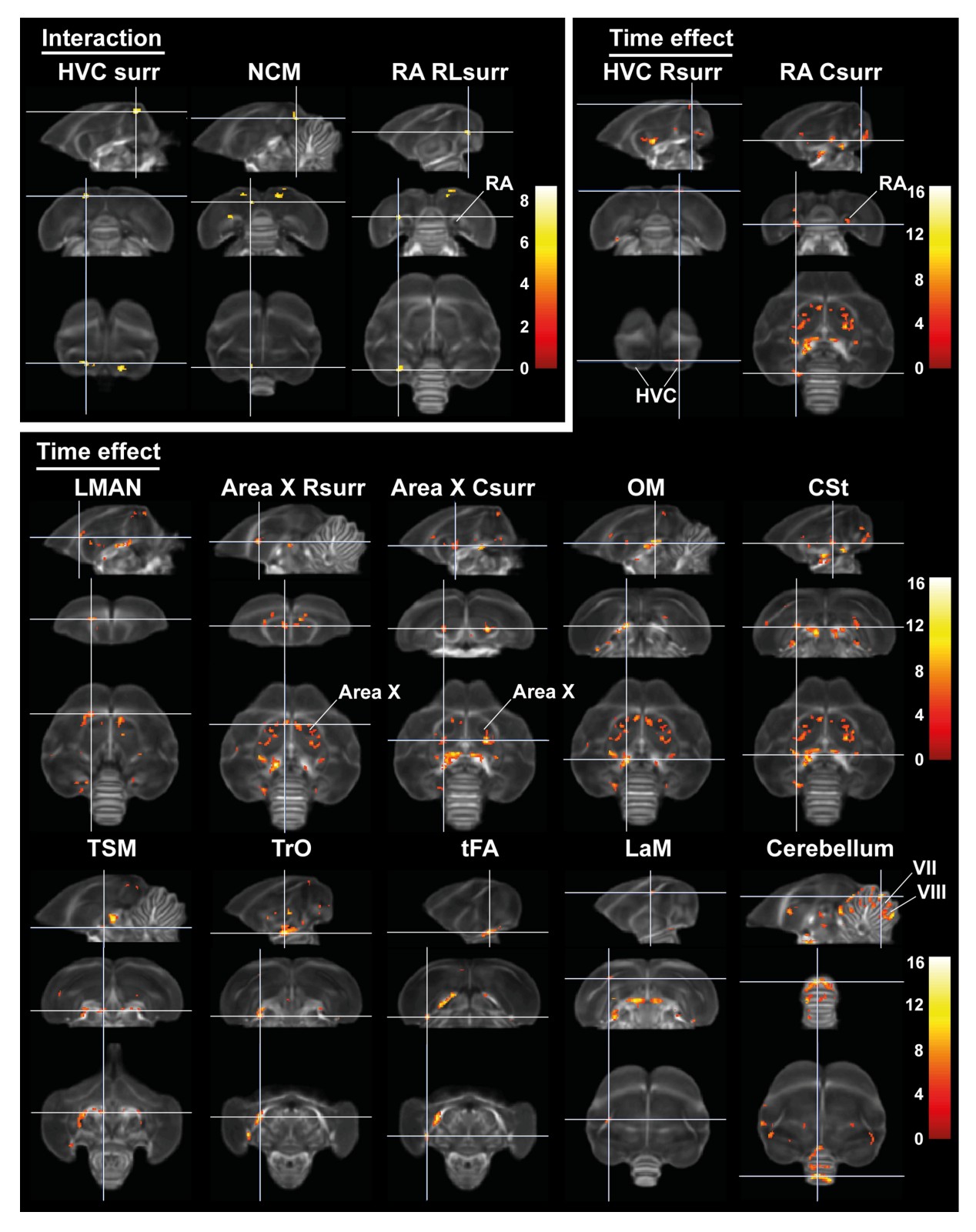

**Figure 7.** Time effect and interaction between sex and time in fractional anisotropy (FA). The statistical maps were assessed at $p_{uncorr} < 0.001$ and $k_E \geq$ 10 voxels with a small volume correction including regions of the white matter tracts, auditory and song control system. Main time effect of the cerebellum cluster was assessed without the small volume correction. Color scale represents significance by F-value. C, caudal; RL, rostro-lateral; R, rostral; surr, surroundings.

**Table 4.** Clusters displaying a significant change over time and the interaction between sex and time in fractional anisotropy (FA). Table summarizing the significance at cluster level and peak level. Some clusters are large and cover multiple regions, indicated by the single statistic at cluster level. For each sub-region within this large cluster, the peak significance (pFWE) and T-value (T) are reported. p-Values are FWE corrected. $K_E$ indicates the number of voxels within a cluster. Gray values indicate clusters that are significant at cluster level but not at peak level.surr, surroundings.

| Effect | Cluster | Hemisphere | Cluster | | Peak | |
|---|---|---|---|---|---|---|
| | | | $p_{FWE}$ | $k_E$ | $p_{FWE}$ | $F$ |
| Interaction time * sex | HVC surr | Left | 0.001 | 61 | 0.013 | 8.75 |
| | | Right | 0.003 | 50 | 0.136 | 7.31 |
| | RA rostro-lateral surr | Left | 0.013 | 36 | 0.139 | 7.29 |
| | NCM | Left | 0.048 | 26 | 0.331 | 6.65 |
| Main time effect | HVC rostral surr | Right | 0.032 | 29 | 0.526 | 6.24 |
| | RA caudal surr | Left | <0.0001 | 86 | 0.009 | 8.93 |
| | RA rostro-medial surr | Right | 0.006 | 42 | 0.017 | 8.57 |
| | Area X caudal surr | Right | <0.0001 | 222 | <0.0001 | 14.96 |
| | | Right | 0.128 | 19 | 0.016 | 8.61 |
| | | Left | <0.0001 | 131 | 0.129 | 7.35 |
| | | Left | 0.148 | 18 | 0.039 | 8.07 |
| | Area X rostral surr | Left | <0.0001 | 76 | 0.007 | 9.09 |
| | | Left | 0.037 | 28 | 0.144 | 7.27 |
| | | Right | <0.0001 | 176 | <0.0001 | 11.05 |
| | LMAN | Left | <0.0001 | 140 | 0.022 | 8.40 |
| | OM | Left | <0.0001 | 513 | <0.0001 | 16.44 |
| | | Right | <0.0001 | 158 | 0.002 | 9.75 |
| | CSt | Left | <0.0001 | 82 | 0.003 | 9.53 |
| | | Right | 0.001 | 61 | 0.048 | 7.95 |
| | | Right | 0.025 | 31 | 0.242 | 6.89 |
| | TSM | Right | 0.009 | 39 | 0.009 | 8.97 |
| | | Left | 0.001 | 56 | 0.074 | 7.70 |
| | tFA | Left | <0.0001 | 91 | <0.0001 | 12.73 |
| | | Right | 0.015 | 35 | 0.042 | 8.03 |
| | TrO | Left | <0.0001 | 302 | <0.0001 | 14.18 |
| | | Right | 0.001 | 58 | 0.264 | 6.83 |
| | LaM | Left | 0.037 | 28 | 0.120 | 7.40 |
| | Cerebellum | Lobule VII | <0.0001 | 1779 | <0.0001 | 15.26 |
| | | Lobule VIII | | | <0.0001 | 13.17 |

based measures from these ROIs to provide more insight into the basis of the fractional anisotropy change (*Figure 8—figure supplements 1–4*). For the statistical analysis of the extracted parameters, we conducted a linear mixed model analysis with Tukey's Honest Significant Difference (HSD) multiple comparison correction during post hoc statistics.

## Sexual dimorphism in seasonal neuroplasticity is lateralized and restricted to specific regions of the song control and auditory system

Voxel-based analysis picked up a significant interaction between sex and time in the surroundings of HVC (bilateral), in a rostro-lateral part of left RA surroundings and in left NCM (*Figures 7* and *8*). In males, fractional anisotropy increases significantly in specific parts of the surroundings of main song control nuclei (HVC and left RA surroundings) and of the auditory system (left NCM), in females

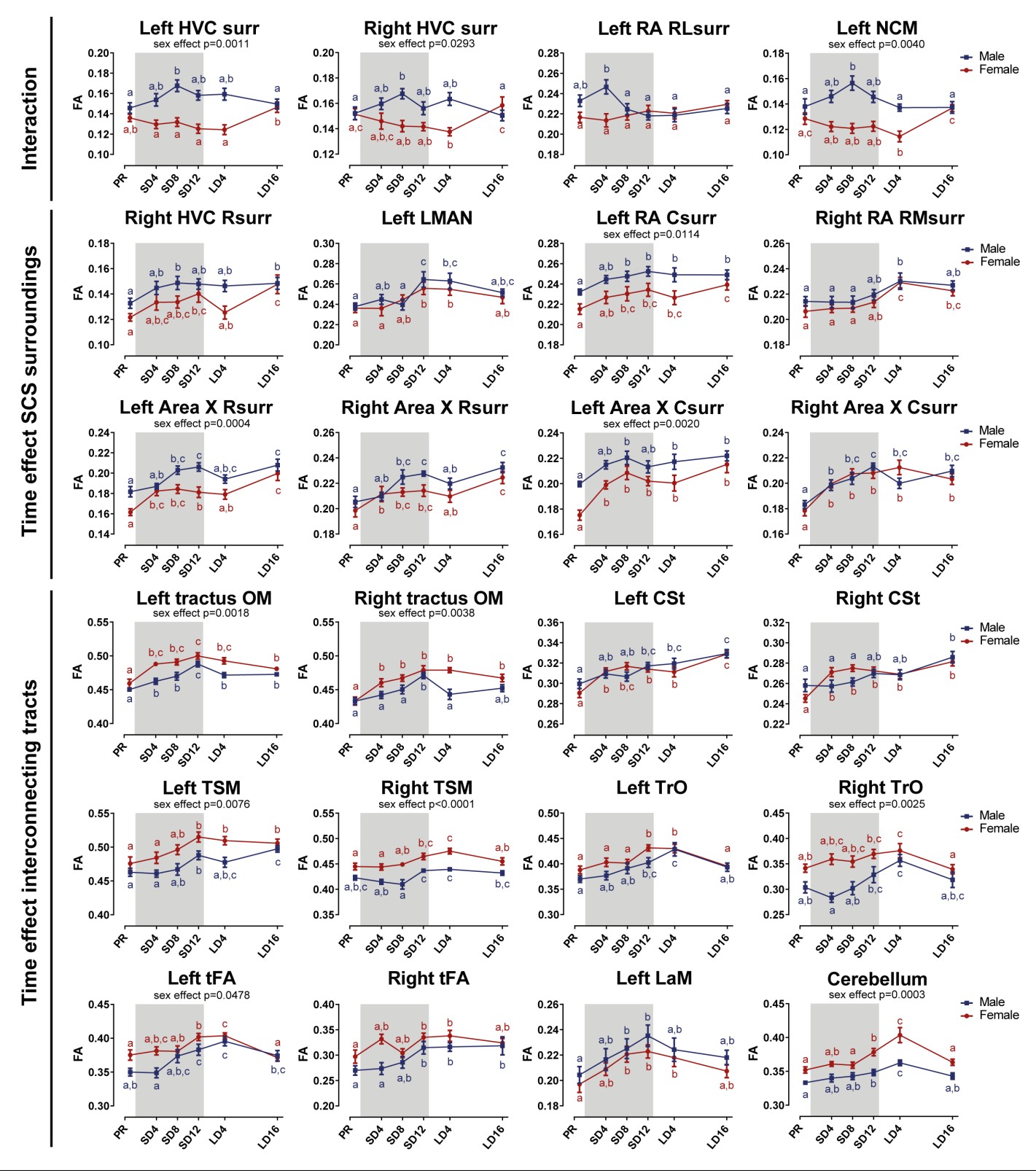

**Figure 8.** Summary of the longitudinal changes over time of fractional anisotropy (FA) extracted from ROI-based clusters that showed a significant interaction (first row), a significant change over time in the surroundings of song control nuclei (row 2 and 3) or a significant change over time in the fiber tracts (row 4–6). The gray area indicates the entire photosensitive period of short days (8L:16D). Significant sex differences are reported with their p-value under the respective ROI-based cluster. Different letters denote significant differences by comparison with each other in post-hoc t-tests with

*Figure 8 continued on next page*

*Figure 8 continued*

p<0.05 (Tukey's HSD correction for multiple comparisons) comparing the different time points to each other. If two time points share the same letter, the fractional anisotropy values are not significantly different from each other. C, caudal; RL, rostro-lateral; R, rostral; surr, surroundings.

The online version of this article includes the following source data and figure supplement(s) for figure 8:

**Source data 1.** This file contains the source data used to make the graphs presented in *Figure 8* and *Figure 8—figure supplement 1–4* .

**Figure supplement 1.** Summary of the longitudinal changes over time in FA, MD, AD, RD, FD, and log FC extracted from ROI-based clusters that had a significant interaction, including HVC surroundings (surr), left RA rostro-lateral surroundings (RLsurr), and NCM.

**Figure supplement 2.** Summary of significant the longitudinal changes over time in FA, MD, AD, RD, FD, and log FC extracted from ROI-based clusters surrounding several song control nuclei including right HVC rostral surroundings (Rsurr), left RA caudal surroundings (Csurr), Area X rostral surroundings, and caudal surroundings.

**Figure supplement 3.** Summary of the significant longitudinal changes over time in FA, MD, AD, RD, FD, and log FC extracted from ROI-based clusters at level of LMAN and several tracts related to song and auditory processing including OM, CSt, and LaM.

**Figure supplement 4.** Summary of the significant longitudinal changes over time in FA, MD, AD, RD, FD, and log FC extracted from ROI-based clusters at level of TrO, TSM, tFA, and cerebellum.

fractional anisotropy values remain constant or even decrease over time. Male fractional anisotropy increase during the photosensitive period in NCM and HVC surroundings is reflected in increased diffusion in all directions (mean, axial, and radial diffusivity increase over time) (*Figure 8—figure supplement 1*).

## Similar seasonal neuroplasticity in both sexes mostly occurs in the photosensitive stage

While some specific regions of the song control and auditory circuit present sexually dimorphic neuroplasticity, most neuroplasticity occurs in parallel in males as in females, not only in regions related to song control but also in the auditory and visual system and the cerebellum (*Figures 7* and *8*). Within the song control system, sexually analogous neuroplasticity occurs in the surroundings of right rostral HVC, RA, Area X (rostral and caudal surroundings) and within left LMAN. Some of the white matter tracts involved in song behavior such as the OM, LaM, and CSt also display neuroplasticity. OM conveys projections from the arcopallium, including connections from RA to downstream regions (*Avian Brain Nomenclature Forum et al., 2004*), whereas CSt holds projections going to the auditory system (*Vates et al., 1996*). LaM is the dividing lamina between the mesopallium and nidopallium and caries connections of HVC to Area X and of LMAN to RA (*Nottebohm et al., 1982*). After 4 weeks of short days, our earliest time point of the photosensitive period, females presented a significant fractional anisotropy change in the OM, rostral, and caudal Area X surroundings and the CSt, whereas males only had significant fractional anisotropy changes in the left OM, left RA surroundings (interaction cluster) and caudal surroundings of right Area X at that time point. Most of the neuroplasticity of song control nuclei like HVC and Area X surroundings in males occurred only after eight weeks of short days. It seems as if the song-related seasonal neuroplasticity occurs slightly sooner in females compared to males, especially in the rostral surroundings of Area X, which is a part of the anterior forebrain pathway, and the CSt, which is an auditory region. Both structures could play a role in the auditory feedback loop that is associated with vocal learning (*Bosíková et al., 2010*). In many song control nuclei, fractional anisotropy values remain high during photostimulated and photorefractory period, only in some parts of the Area X and right HVC surroundings females show a slight dip in fractional anisotropy during the photostimulated phase, before increasing again during the photorefractory phase. The LMAN is one of the last song control nuclei to change with increased fractional anisotropy levels from SD12 onward.

Furthermore, many of the seasonal changes in brain structures related to the singing behavior do not return to baseline levels after sixteen weeks of long days. In males, the surroundings of HVC, left RA, and left NCM were not different between the two sample points in the photorefractory phases (*Figure 8—figure supplement 1*). Some structures such as the left OM show a gradual decrease in fractional anisotropy values at LD16 compared to their peak at SD12. A potential explanation is that these singing-related structures require more time to return to baseline levels. Alternatively, certain regions related to song behavior may never fully regress to baseline photorefractory levels, but instead increase year after year. *Bernard et al., 1996* demonstrated that in yearling starlings, HVC and RA volume was lower compared to older adult starlings, which was linked to the increasing song

repertoire of starlings over the years. Furthermore, CSt related to the auditory system distinguishes itself from other structures as it keeps on increasing in fractional anisotropy over subsequent photoperiods. This could represent a maturation upon aging, independent of the seasonal changes.

In other white matter structures, such as the TrO, TSM, and tFA; most neuroplastic changes in fractional anisotropy and fiber density start during the photosensitive phase around SD12 (*Figure 8— figure supplement 4*). As the name indicates, the optic tract is involved in visual processing. TSM is also involved in the visual pathway as it conveys projections from the wulst to sites within the brain stem (*Avian Brain Nomenclature Forum et al., 2004*). tFA holds projections of the trigeminal sensorimotor pathway between the basorostral nucleus (in the rostral pallium) and arcopallium (*Avian Brain Nomenclature Forum et al., 2004*; *Wild and Farabaugh, 1996*). Neuroplasticity of the visual system could be relevant to prepare the birds for the breeding season, where visual cues like ultraviolet plumage colors are important for mate selection (*Bennett et al., 1997*). Interestingly, cerebellar lobules VII-VIII, which are part of the oculomotor cerebellum, follows a similar temporal profile of neuroplasticity as other cerebral structures, with significantly increased fractional anisotropy values by the end of the photosensitive period. This shows that multisensory neuroplasticity is not limited to the cerebrum, but also involves the cerebellum, something that has not yet been observed in songbirds.

Male neuroplastic changes in fractional anisotropy are mostly attributed to changes in axial diffusivity, sensitive to changes in axon number or axon density (*Figure 8—figure supplements 2– 3*). Female starlings conversely demonstrated decreased radial diffusivity during the photosensitive period in right HVC, Area X surroundings, LaM and cerebellar lobule VII-VIII, indicating changes in myelin content or axon diameter, which was minimal in males (*Figure 8—figure supplements 2–4*).

In line with the voxel-based analysis for sex difference, the hemispheric asymmetry in males and increased interhemispheric connectivity is also reflected in the extracted diffusion parameters of these clusters. Female starlings have higher fractional anisotropy and fiber density in tracts like OM and CSt compared to male starlings (*Figure 8*, *Figure 8—figure supplement 3*). Males present higher fractional anisotropy in clusters of the left hemisphere, like NCM, RA and Area X surroundings. However, both left and right Area X surroundings of males have lower diffusion in all directions (mean, axial and radial diffusivity) compared to females, suggesting a higher tissue density or fiber organization in males (*Figure 8—figure supplement 2*). Both left and right RA surroundings have lower radial diffusivity in males compared to females, suggesting males have more myelin content or smaller axon diameter (see *Table 1*). However, in the right RA surroundings, axial diffusivity is also significantly lower in males, suggesting a lower axon number or axon density.

Overall, neuroplasticity in structures related to song and auditory processing starts early into the photosensitive period (SD4 and SD8). Structures related to other sensory systems, such as the TrO, TSM, tFA, and cerebellum, only show significant increases by the end of the photosensitive stage (SD12) and often have again decreased fractional anisotropy values by the next photorefractory phase. Even though the majority of these neuroplastic changes occur similarly in males and females, they sometimes differ in their approach. Fractional anisotropy changes in males are mostly due to changes in axial diffusivity, however females show changes in radial diffusivity over time.

## Photoperiodic variation in hormone levels and song behavior

In order to determine how the changes in neuroplasticity are related to changes in hormones and singing behavior across the different photoperiods, we collected blood samples and recorded songs at each time point. A linear mixed model analysis was performed to determine differences between sex, time and identify their interaction with HSD multiple comparison correction. This way, we validated whether the artificial photoperiod schedule induced hormonal changes comparable to the natural seasonal gonadal cycle in starlings.

### Hormone levels

After 4 weeks of photostimulation, male testosterone levels increased to (1.67 ± 0.24) ng/ml (M ± SE) (*Figure 9A*). Two out of the 13 male starlings already showed elevated plasma testosterone levels above 1 ng/ml at the end of the photosensitive phase, but this increase was not significant at group level. In females, plasma testosterone levels are significantly elevated at the end of the

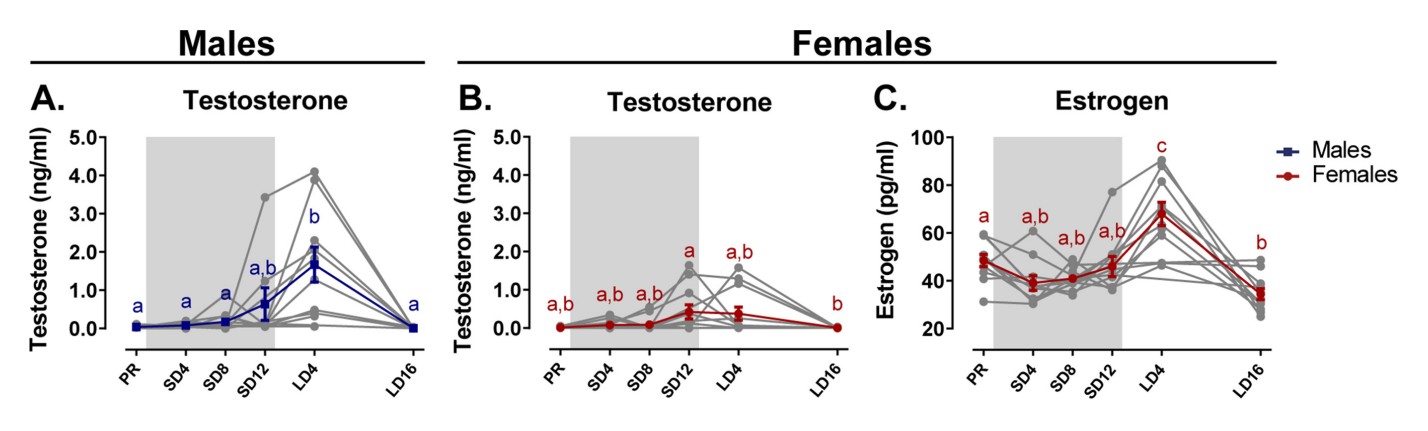

**Figure 9.** Overview of the seasonal changes in plasma testosterone (A, B) and estrogen (C) concentrations in male and female starlings. Gray lines depict individual animals, while blue and red lines represent the group average with the standard errors of the mean error bars. The gray area indicates the photosensitive period of short days (8L:16D). Different letters denote significant differences by comparison with each other in post-hoc t-tests with p<0.05 (Tukey's HSD correction for multiple comparisons) comparing the different time points to each other. If two time points share the same letter, the testosterone level or estrogen level are not significantly different from each other.

The online version of this article includes the following source data for figure 9:

**Source data 1.** This file contains the source data used to make the graphs presented in *Figure 9*.

photosensitive phase SD12 (0.43 ± 0.11) ng/ml (*Figure 9B*). Even though both sexes increase their circulating testosterone levels, female starlings do not reach the same levels as male starlings.

Since testosterone is not the main reproductive hormone in female starlings, we also looked at changes in circulating estrogen levels (*Figure 9C*). Similar to male testosterone levels, female estrogen levels peak after 4 weeks of photostimulation (M = 67.8, SE = 3.26). These increased circulating hormones at photostimulation confirm the reproductive maturation associated with the breeding season in starlings. Furthermore, we show that circulating testosterone levels already gradually increase during the photosensitive period.

## Seasonal variation in song behavior and the role of brain size

Under artificial conditions, male starlings are consistently observed to sing at every photoperiod. In contrast, female starlings do show seasonal variation in song rate. They sang most during the photosensitive phase (*Figure 10A*). During the photostimulated phase, the song rate in females is drastically lower compared to males. In female starlings, the song bout length increased gradually over time from (22.7 ± 1.5) s (M ± SE) at the photorefractory phase to (28.6 ± 0.8) s at the end of the photosensitive phase, when their song rate and testosterone levels were the highest (*Figure 10B*). Male starlings presented a similar increase in song bout length over the sequential photoperiods from (24.3 ± 1.3) s at SD4 to (33.2 ± 1.1) s in the photostimulated phase. Interestingly, song bout length was positively correlated at the within-subject level to testosterone plasma levels in male starlings (rmcorr r = 0.632, p=0.037). Such a repeated measures correlation was not present in female starlings; neither did the song rate correlate with testosterone levels in either sex.

We further investigated the effect of brain size on song behavior, using brain size (defined in result section Role of brain size in the general sexual dimorphism) as a fixed factor in the analysis of the song behavior for each sex separately. Interestingly, the song rate of female starlings presented a significant effect of brain size and interaction between brain size and time respectively (F(1, 10.8)=5.74, p=0.0358; F(5, 37.6)=3.14, p=0.0181). Post hoc analysis quickly revealed that female starlings with a small brain rarely sing during any of the time points (*Figure 10C*). Large brain females were most responsible in determining the seasonal change in song rate and song bout length. Further analysis revealed that the brain size of the female birds is proportional to their body weight, evident by positive Pearson's correlation between the averaged brain size and averaged body weight, which is not the case in the male starlings (*Figure 10D*). Female starlings with a large brain have a higher body weight and are capable of singing with a high song rate during the photosensitive period

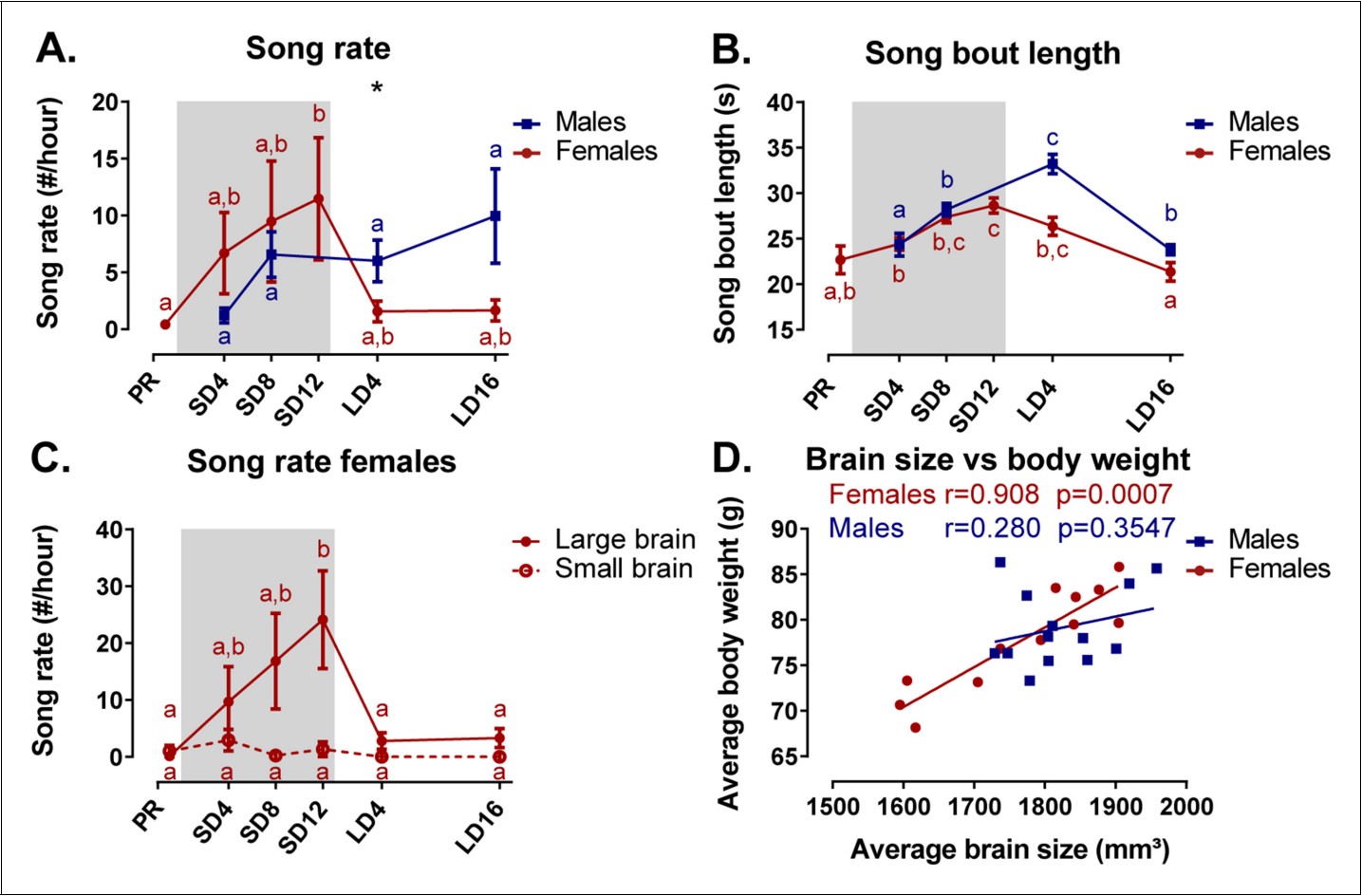

**Figure 10.** Overview of the seasonal changes in song rate (A) and song bout length (B) in male and female starlings. Dividing the starlings based on brain size showed that only large brain female starlings sing (C). Females present a significant correlation between the average brain size and body weight (D). (A–C) Blue and red lines represent the group average with the standard errors of the mean error bars. The gray area indicates the photosensitive period of short days (8L:16D). Different letters denote significant differences by comparison with each other in post-hoc t-tests with p<0.05 (Tukey's HSD correction for multiple comparisons) comparing the different time points to each other. If two time points share the same letter, the song rate or song bout length are not significantly different from each other. (D) The Pearson's correlation and its significance are reported for males and females.

The online version of this article includes the following source data for figure 10:

**Source data 1.** This file contains the source data used to make the graphs presented in *Figure 10*.

unlike their smaller female counterparts. Importantly, the interpretation of the sexual dimorphisms attributed to brain size is further complicated, since the difference between large and small brain female starlings is confounded by their difference in singing behavior.

## Neural correlates of song behavior

Having established the seasonal changes in neuroplasticity and song behavior, we performed a voxel-based multiple regression analysis to determine the neural correlate of song behavior. Neither song bout length nor the male song rate presented a significant voxel-based correlation. However, female song rate correlated positively to fractional anisotropy in the left hippocampus and the surrounding fibers of RA and Area X (*Figure 11*, *Table 5*). This correlation was not only linked to the overall difference between singing and non-singing females, but also to the increase of song rate within a subject as evidenced by the significant repeated measures correlation with fractional anisotropy of the caudal surroundings of Area X and left hippocampus. Thus, we conclude that not only are large female starlings capable of producing more songs, this increased song rate is also

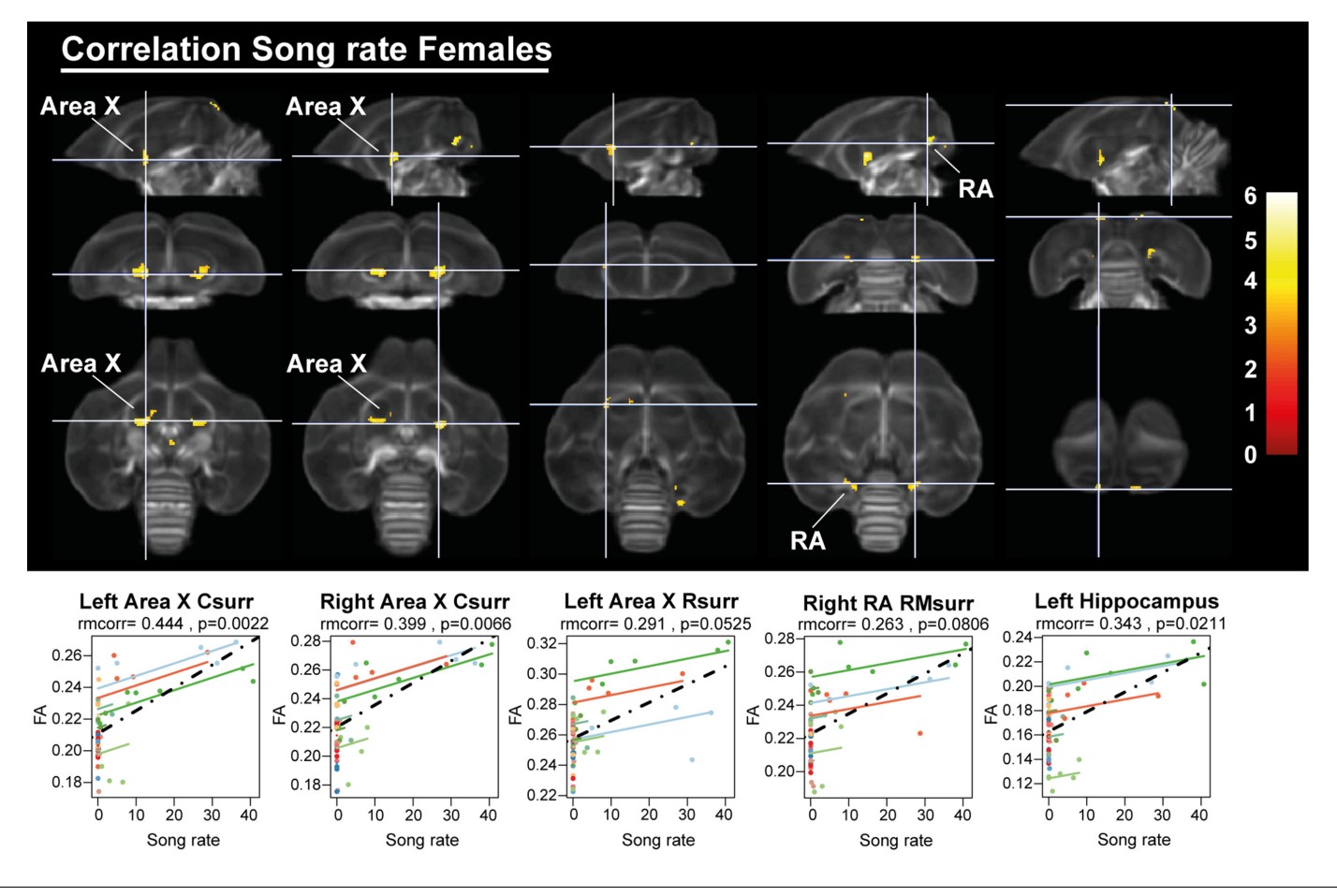

**Figure 11.** Overview of structural neural correlates of song rate to fractional anisotropy in female starlings. The statistical maps were assessed at $p_{uncorr}$ <0.001 and $k_E \geq 10$ voxels with a small volume correction including regions of the song control system and other white matter structures. Below each statistical parametric map, the identified correlations were further explored with repeated measures correlation. Solid colored lines show the best linear fit for the within-subject correlation using parallel regression lines for individual animals. The dashed line represents the linear fit of the overall Pearson correlation representing the between-subject correlations. C, caudal; RM, rostro-medial; R, rostral; surr, surroundings.

The online version of this article includes the following source data for figure 11:

**Source data 1.** This file contains the source data used to make the graphs presented in *Figure 11*.

correlated to improved microstructural organization of some major song control nuclei and the hippocampus.

## Discussion

The present data-driven longitudinal study confirms the well-established sexual dimorphism of the songbird brain, showing that male starlings generally have a more structured song control system compared to female starlings- as is the case in zebra finches where only the male sings (*Hamaide et al., 2017*). These data also lead to new findings, including the observation that females have stronger tracts interconnecting hemispheres and nuclei, while males have a microstructural asymmetry toward the left hemisphere. Interestingly, brain size does not completely explain the sexually dimorphic hemispheric asymmetry of males or increased interconnectivity of female brains; only some of the sex differences in certain song control nuclei could be attributed to brain size. However, the difference in brain size is confounded by the difference in singing behavior of female starlings. Female starlings with small brains rarely sing during any of the time points. In contrast, females with

**Table 5.** Summary of the voxel-based correlation analysis of song rate vs fractional anisotropy.

This table summarizes for each significant cluster-based ROI the voxel-based FWE corrected p-value at cluster and peak level, next to the overall Pearson's correlation, the repeated measures correlation (rmcorr) and its significance of the averaged fractional anisotropy values versus the respective song characteristic. $K_E$ indicates the cluster size. T reflects the t-value of the maximal peak located within the cluster-based ROI. Voxel-based correlations in gray are significant at the cluster level, but not at peak level. *Significant repeated measures correlations are indicated in bold.* surr, surroundings.

| Cluster | Hemi-sphere | Cluster | | Peak | | Pearson's correlation | Rmcorr | |
|---|---|---|---|---|---|---|---|---|
| | | $p_{FWE}$ | $k_E$ | $p_{FWE}$ | $T$ | $r$ | $r$ | $P$ |
| Area X caudal surr | Left | <0.0001 | 168 | 0.002 | 6.05 | 0.5396 | **0.444** | **0.0023** |
| | Right | 0.001 | 116 | 0.015 | 5.45 | 0.5045 | **0.399** | **0.0066** |
| Area X rostral surr | Left | 0.041 | 45 | 0.427 | 4.24 | 0.4587 | 0.291 | 0.0525 |
| RA rostro-medial surr | Right | 0.015 | 59 | 0.039 | 5.13 | 0.3969 | 0.263 | 0.0806 |
| Hippocampus | Left | 0.132 | 30 | 0.029 | 5.22 | 0.3883 | **0.343** | **0.0211** |

large brains are capable of singing at a high song rate and strengthen their song control network during the photosensitive period. In general, both male and female starlings display similar structural neuroplasticity not only in the surrounding fibers of the song control system, but also in tracts related to other sensory systems such as the auditory and visual network. Even specific parts of the cerebellum, related to the processing of sensory information, display seasonal neuroplasticity. However, males and females show slight differences in the way neuroplasticity is displayed. Male starlings, for example, demonstrated significant neuroplasticity in HVC surroundings and left NCM, which was absent in female starlings. Importantly, most of these neuroplastic changes already start during the photosensitive period, when circulating testosterone levels are low but gradually begin to increase in both sexes.

## General sexual dimorphism in the starling brain's tractogram

In vivo DTI and fixel-based analysis could capture the well-established structural sexual dimorphism at the level of the song control system in songbirds (*Bernard et al., 1993*; *Hamaide et al., 2017*; *Nottebohm and Arnold, 1976*). Even though the sexual dimorphism in the song control system is bilateral, we demonstrated a strong asymmetry toward the left nidopallium and mesopallium in male starlings. Furthermore, analysis of seasonal neuroplasticity confirmed this lateralization, as many clusters found in the left hemisphere (i.e. surroundings of left HVC, RA, rostral and caudal surroundings of Area X) had higher fractional anisotropy values in males compared to females. In general, fractional anisotropy provides a correlate for a set of underlying microstructural characteristics related to structural and spatial organization of axons, dendrites and even cells. Other diffusion parameters can help the interpretation but cannot pinpoint a single underlying cellular event (*Sagi et al., 2012*; *Zatorre et al., 2012*). However, our observations might guide histological studies to shed more light on the underlying structural sexual dimorphisms. Left-right differences are not always examined in histological studies. Most of the studies concerning hemispheric asymmetry or lateralization focus on the lateralization of functions involved in the processing of auditory, visual, and olfactory information (*Rogers, 2012*). Often these studies use electrophysiology or functional MRI (fMRI) to quantify the neural response to certain stimuli (*De Groof et al., 2017*; *Yang and Vicario, 2015*). Human imaging studies have combined fMRI and tract-tracings and found that white matter asymmetries may be one of the factors underlying functional hemispheric asymmetries (*Ocklenburg et al., 2013*). Our study contributes to this hypothesis, showing a large microstructural hemispheric asymmetry of the male starling brain, which might be underlying some of the functional hemispheric asymmetries in starlings. Furthermore, this lateralization might be species-specific; zebra finches by contrast exhibit no left-right difference in the volume of song control nuclei (*Nixdorf-Bergweiler, 1996*), nor did prior DTI studies in zebra finches report a sex difference in cerebral asymmetry of song control nuclei (*Hamaide et al., 2017*).

Unexpectedly, females have higher fractional anisotropy in several tracts interconnecting nuclei of the visual and song control system and in interhemispheric connections such as the commissure

anterior and posterior. This is in contrast to zebra finches where males have a larger white matter volume compartment compared to females (*Hamaide et al., 2018a*). However, stronger interhemispheric connectivity is present in female mammals, although this has been attributed to brain size rather than a real effect of sex (*Hänggi et al., 2014*). Even though male starlings had a slightly -but not significantly- larger brain compared to females, matching our findings in zebra finches (*Hamaide et al., 2018a*), our study could not confirm the hypothesis of neuronal interconnectivity (*Hänggi et al., 2014*), which states that smaller brains are stronger connected interhemispherically, while larger brains exhibit more hemispheric asymmetry caused by relatively stronger interhemispheric connectivity. The voxel-based analysis for brain size could not explain the hemispheric asymmetry in males or the increased interhemispheric connections in females. Brain size could only explain some sexual dimorphisms in specific parts of the song control system, such as the caudal surroundings of the right Area X, surroundings of RA and HVC. However, in females, the difference in brain size is confounded by the difference in song behavior. Our data reveal that female starlings with a large brain sing more, especially during the photosensitive phase, compared to female starlings with a relatively smaller brain. Furthermore, female song rate is correlated to fractional anisotropy in the same song control regions, for example the surroundings of Area X and RA. Brain size is affected by multiple factors including age, circulating hormone levels, environment and experience (*Healy and Rowe, 2007*). Of note, because we used wild caught starlings we cannot control for their exact age or developmental history. The larger females could be older compared to the smaller female starlings. In summation, these confounding factors make it unlikely that the brain size difference is the sole explanation for the sexual dimorphisms.

Sexual structural dimorphisms of the brain might also be explained by sexual differences in behavior. Take, for example, the sexual dimorphism of the TrO and TSM that carries projections to the wulst (*Avian Brain Nomenclature Forum et al., 2004*). Enhancement of the visual system in females could be related to the visual cues like ultraviolet plumage colors female starlings take into consideration during mate selection (*Bennett et al., 1997*). Moreover, this role of the visual system during the breeding season is further corroborated by the neuroplastic changes in the visual system, which occur at the end of the photosensitive period, right before the onset of the photostimulated period and return to baseline levels after sixteen weeks of long days. This is consistent with one of our prior DTI studies, where the optic chiasm of male starlings had lower fractional anisotropy values during the photorefractory state compared to the breeding season (*De Groof et al., 2008*).

Interestingly, because of the limited research in the cerebellum of songbirds, we are the first to report clear sexual dimorphisms in the fiber characteristics of specific lobules of the cerebellum. Females had higher fractional anisotropy values in a large part of the cerebellum covering both anterior and posterior lobules IV-VIII, and higher fiber density in anterior lobules II-IV. In males, mostly the posterior lobule VII had higher fractional anisotropy, whilst lobule VIII had higher fiber density and fiber-bundle cross-section. The avian cerebellum has a strict topographic organization where each cerebellar lobule has a distinct input and output (*Arends and Zeigler, 1991*). Anterior lobules (I-V) contain the somatotopic representation of the tail, leg, and wings (*Whitlock, 1952*) and are more pronounced in avian species with strong hind limbs (*Iwaniuk et al., 2007*). Cerebellar lobules VII and VIII are part of the oculomotor cerebellum (VI-VIII) and receive visual input (*Clarke, 1977*), input from the pretecto-ponto-cerebellar system involved in flight (*Wylie et al., 2018*) and have a somatotopic representation of the cochlea and retina (*Whitlock, 1952*). The pronouncement of the oculomotor cerebellum in males versus females could reflect the integration of visual and auditory sensory input, including the more developed song behavior of males. Language tasks in humans mostly involve the lateral lobules VI and crus I/II, which are implicated in initiating the motor sequence of phonological content, whereas VIIb/VIII support the phonological storage during the maintenance of verbal information (*Mariën et al., 2014*). However, there is only limited information on the specific cerebellar lobules implicated in song production of songbirds. Recently a cerebello-thalamic-basal ganglia pathway was identified that influences song processing, where lateral cerebellar nucleus (CbL) projects to dorsal thalamic regions and coincides with retrograde labeling of Area X and medial striatum (*Person et al., 2008*; *Pidoux et al., 2018*). Furthermore, lesioning Area X caused structural remodeling within the cerebello-thalamo-basal ganglia pathway including the dorsal thalamic zone, CbL and extended further in cerebellar lobules VIb and VII (*Hamaide et al., 2018b*). These lesions independently affected the duration of syllables (*Hamaide et al., 2018b*; *Pidoux et al., 2018*), which indicates the importance of the cerebello-thalamo-basal ganglia pathway

in the timing of birdsong (*Konopka and Roberts, 2016*). The cerebellum generally works as a forward controller, where it compares intention with execution and uses the error information to correct future events (*D'Angelo, 2018*; *Strick et al., 2009*). This conceptualization of the cerebellum as a learning machine was developed as a model for motor control (*Miall and Reckess, 2002*), but can also be applied in other learning behaviors, such as vocal learning in birds, since the forward control process of the cerebellum is especially implicated in the timing of combined stimuli. Moreover, the oculomotor cerebellum exhibits neuroplasticity with a similar temporal profile as the multisensory neuroplasticity observed in the cerebrum. The molecular layer of the cerebellum contains the parallel fibers emitted by the granule cells and is known to maintain a high level of neuroplasticity throughout life (*Ito, 2006*; *Medina et al., 2002*). Furthermore, steroids can affect cerebellar Purkinje firing related to locomotor activity (*Dean and McCarthy, 2008*; *Smith, 1995*). Together, these findings imply that the plasticity in song control system is not limited to the known song circuitry but also involves the cerebellum.

The general sexual dimorphism in the starling brain extends well beyond the song control system, and whilst for mammals this was suggested to be related to the difference in brain size, we do not find evidence for that in avian species. The few sex differences in the song control system that could be attributed to brain size are confounded by song behavior. Furthermore, many of the other sexual dimorphisms involving the visual system and cerebellum could be explained by behavioral differences between sexes. It can thus be reasonably assumed that the general structural sexual dimorphisms in starlings are related to differences in behavior rather than brain size.

## Seasonal variation in gonadal hormones song behavior and its Neural correlates

The seasonal variation in hormone levels (*Dawson, 1983*; *Riters et al., 2002*; *Williams et al., 2004*) and song behavior during artificial photoperiod switching is in good agreement with the seasonal changes observed in natural conditions. In male starlings on a natural cycle, hypothalamic GnRH expression is already high in February under short days, which coincides with the start of the testes growth and results in a gradual increase of circulating testosterone (*Bentley et al., 2013*; *Dawson, 1983*). Female birds also produce testosterone, mainly in the ovaries and the adrenal gland (*Tanabe et al., 1979*). Circulating testosterone levels are high in female starlings in the pre-laying phase and decrease after ovulation, whereas estrogens are high during the breeding season in March and peak around ovulation (*Dawson, 1983*; *Williams et al., 2004*). It is well known that estrogens are fundamental in the regulation of female reproduction, whereas the role of testosterone in female reproduction is less studied. However, female starlings are most aggressive during the prelaying period, when their testosterone levels are high (*Williams et al., 2004*). Our study confirms that both male and females present a gradual increase in circulating testosterone during the late photosensitive phase. In males, this was shown to coincide with an increase in hypothalamic GnRH expression (*Bentley et al., 2013*). Furthermore, the current study showed that male testosterone plasma levels were positively correlated to the song bout length on a within-subject level. We suggest that beside the effect of testosterone on reproductive functions, it could also be important in the regulation of neuroplasticity observed during the photosensitive phase.

## Male and female starlings both experience neuroplasticity in their song control system, but display differential underlying mechanisms and song behavior

Although structural neuroplasticity of the song control system is broadly similar in males and females, there are some notable differences. We revealed that seasonal neuroplasticity differs between males and females in specific parts of the HVC surroundings and left NCM, where only males experienced seasonal neuroplasticity. Females by contrast only showed a fractional anisotropy increase in the more rostral part of the right HVC surroundings after 12 weeks of short days. During photostimulation, when the female singing behavior becomes less, fractional anisotropy values in the right HVC surroundings dropped slightly. In other structures related to song and auditory processing, neuroplasticity starts early into the photosensitive period (SD4 and SD8) in both males and females. Furthermore, fractional anisotropy values in the Area X surroundings and hippocampus were positively correlated to the female song rate on a within-subject level. The hippocampus

receives input from various parts of the pallium and dorsal thalamus that process somatosensory, olfactory, visual, and auditory information (*Striedter, 2016*). The role of hippocampus in song behavior is not widely understood. Previous work has mainly focused on the role of the hippocampus in spatial memory performance (*Macphail, 2002*). Prior studies demonstrated that males generally outperform females in spatial memory performance tasks, but rely on different strategies in learning such cognitive tasks in both mammals and songbirds (*Kosarussavadi et al., 2017*; *Rensel et al., 2015*; *Tropp and Markus, 2001*). Sexually different learning mechanisms have also been suggested in vocal learning in songbirds (*Riebel, 2016*). Few studies have looked at the role of the hippocampus in song behavior in zebra finches. Hippocampal lesions during song learning and adulthood in male zebra finches did not affect song learning, singing or song structure. However, in females, hippocampal lesions affected preference for their father's song (*Bailey et al., 2009*). Interestingly, the hippocampus has a high aromatase expression and is a site of immediate early gene expression in response to conspecific zebra finch song, predominantly in female zebra finches (*Bailey and Saldanha, 2015*; *Bailey and Wade, 2003*; *Gobes et al., 2009*). These findings suggest that the hippocampus might play a role in specific characteristics of song perception, like identity and temporal context. To our knowledge, we are the first to report on a potential role of the hippocampus in the song behavior of female starlings.

Together these findings suggest that males and females use different vocal learning strategies relying on different parts of the song control system and displaying differential neuroplasticity mechanisms. Female starlings relied particularly on the auditory feedback loop and showed involvement of the hippocampus. Males, on the other hand, display greater activity-induced neuroplasticity in the NCM, left RA and HVC surroundings.

## The photosensitive phase is a sensitive window where both sexes experience multisensory neuroplasticity

One of the most important findings of this study is that most of the neuroplastic changes start during the photosensitive phase. Despite widespread structural sexual dimorphism, the majority of the structural seasonal neuroplasticity occurs similarly in males and females. A second major observation is that besides the SCS, various sensory systems and the cerebellum display neuroplasticity in both sexes. This finding could only be partially confirmed with a prior histological study that reported volume changes in song control nuclei during the photosensitive phase of male starlings (*Riters et al., 2002*) as immunohistochemistry is invasive and does not allow dynamic observations of changes in the same animal. These changes in neuroplasticity are a circannual observation driven by the shortening day length, as is the case for several other behaviors and mechanisms in different species (*Stevenson et al., 2015*). The exact mechanism behind these circannual rhythms remains unclear. Of note is that during the photosensitive period, aromatase expression is increased and 5β-reductase (which inactivates testosterone) is decreased in the telencephalon (*Riters et al., 2001*). This indicates that local changes in testosterone metabolism in the brain, could contribute to singing behavior and the observed neuroplasticity during the photosensitive period, even though circulating testosterone levels are still low. In line with male starlings, male photosensitive canaries' HVC volume was not significantly different from photostimulated males, which the authors related to a gradual increase in GnRH expression in the POA during the photosensitive period (*Hurley et al., 2008*).

Just like starlings, canaries are open-ended learners. However, canaries modify their song between breeding seasons (when their testosterone levels are low); once they reach a stable song it remains stable for the rest of the breeding season (*Nottebohm et al., 1986*). Recently, it was shown that this song stabilization is associated with an increase in the perineuronal nets (PNN) expression in RA and Area X of adult male canaries prior to the breeding season (*Cornez et al., 2020*). They hypothesized that the changes in PNN expression were associated with the closing and re-opening of a sensitive period for vocal learning. We hypothesize that a similar pattern of closing and re-opening of sensitive windows exists in starlings. Among open-ended learners, starlings are more plastic compared to canaries, that is, they can learn new song elements throughout the year and can incorporate them in their song up to eighteen months after first hearing them (*Böhner et al., 1990*; *Marler et al., 1994*). However, these experiences only result in long-lasting structural changes in the brain during the photosensitive period, persisting during the photostimulated phase and dissipating in the photorefractory state. As in canaries, male starlings have higher percentage of PNN surrounding PV in the HVC during the late photosensitive and photostimulated phase compared to the

photorefractory state, the early photosensitive stage was not investigated (*Cornez et al., 2017*). This indicates that PNNs could play a role in closing sensitive periods. However, compared to other open-ended learners (canaries) and closed-ended learners (zebra finches), starlings exhibit far less PNN expression in their song control system (*Cornez et al., 2017*). This could be linked to the starling's ability to change its repertoire throughout the year. The relatively low PNN density in the starling brain does not necessarily mean that they cannot use other mechanisms to end their sensitive period. Myelination is another relevant candidate for brain circuitry consolidation when a sensitive window closes (*Knudsen, 2004*; *Xin and Chan, 2020*). Using in vivo DTI, which is sensitive to changes in myelination, we demonstrated that both sexes experienced differential and parallel heightened multisensory neuroplasticity in auditory and visual system and cerebellum mainly during the photosensitive period. This often-neglected period in which gonadal hormones are still very low might represent a 'sensitive window', during which permissive circumstances for activity-induced neuroplasticity re-occur and different sensory and motor systems in both cerebrum and cerebellum are seasonally re-shaped. These findings open new vistas for investigating reopening of sensitive or critical periods of neuroplasticity.

# Materials and methods

## Key resources table

| Reagent type (species) or resource | Designation | Source or reference | Identifiers | Additional information |
|---|---|---|---|---|
| Strain, strain background (Sturnus vulgaris, male and female) | Starlings | Wild caught | | See Materials and methods, Section 1 |
| Commercial assay or kit | estradiol RIA | MP Biomedicals, Solon, Ohio | RRID:SCR_013308 SKU: 0723810-CF | |
| Commercial assay or kit | testosterone RIA | MP Biomedicals, Solon, Ohio | RRID:SCR_013308 SKU: 0718910-CF | |
| Software, algorithm | Raven Pro | Cornell Lab of Ornithology, Ithaca, NY | | Version 1.5 |
| Software, algorithm | MrTrix3 | MrTrix3 (https://www.mrtrix.org) | RRID:SCR_006971 | version 3.0 |
| Software, algorithm | SPM | SPM (https://www.fil.ion.ucl.ac.uk/spm) | RRID:SCR_007037 | Version 12 |
| Software, algorithm | Rmcorr | *Bakdash and Marusich, 2017* 10.3389/fpsyg.2017.00456 | | Version 0.4.3 |
| Software, algorithm | JMP Pro 15 | JMP (https://www.jmp.com) | RRID:SCR_014242 | Version 15.1 |
| Software, algorithm | GraphPad prism | GraphPad Prism (https://graphpad.com) | RRID:SCR_002798 | Version 6 |

## Subjects and experimental setup

All experiments were performed on 13 male and 12 female starlings (*Sturnus vulgaris*). The males were wild caught as adults in Normandy (France) in November 2014 and the females were caught in Nicosia (Cyprus) in March 2017. All subjects were housed in large indoor aviaries (L x W x H: 2.2 × 1.4×2.1 m) at the University of Antwerp with food and water available ad libitum and an artificial light-dark cycle mimicking the natural photoperiod. All birds were kept on a long day photoperiod (16 hr light: 8 hr dark, 16L:8D) in order to induce photorefractory state at the start of experiment. When all birds molted and were fully photorefractory, a first baseline measurement was made. This was in May 2015 for the male starlings, and December 2017 for the female starlings. Both males and females followed the same photoperiodic regime: after the photorefractory baseline scan, they were switched from long to short (8L:16D) days to induce photosensitivity. After 12 weeks of short days, the photosensitive starlings were switched back to long days (16L:8D) to induce photostimulation. They remained on long days, so that they became photorefractory once more. We monitored neuroplasticity repeatedly at six different time points. The first time point was at the end of the photorefractory state (PR). After switching to short days, we measured every four weeks to follow up the

seasonal plasticity during the photosensitive period (SD4, SD8, SD12). Additionally, we measured after four weeks on long days (LD4), when starlings were fully photostimulated. Finally, we measured after sixteen weeks on long days (LD16), when starlings are photorefractory again. At each time point, birds were weighed, songs were recorded, blood samples were taken, and MRI (DTI and 3D scans) was acquired. The housing and experimental procedures were performed in agreement with the European directives, Belgian and Flemish laws and were approved by the Committee on Animal Care and Use of the University of Antwerp, Belgium (2014–52).

## Testosterone and estrogen assay

Blood samples were collected to quantify plasma testosterone concentration in all birds. In female starlings, we also assayed estradiol in addition to testosterone, since this is one of the primary female sex hormones. At each time point, blood collection occurred within a 2 hr window at noon between 11:30 hr and 13:00 hr in order to limit variation of plasma levels due to a daily rhythm. Starlings were housed individually for song recording, which allowed fast capture and blood sampling (within 5 min. of capture) to minimize the increase in testosterone as a result of acute stress (*Van Hout et al., 2010*). The alar wing vein was punctured with a 25-gauge needle to collect 300–500 µl of blood into heparin-coated capillary tubes. The blood was transferred to tubes and centrifuged for 10 min at 2060 g while cooling to 4°C. Plasma was collected and stored at −20°C prior to assay. Estradiol and testosterone concentrations were quantified by Radioimmunoassay (RIA) using a commercial double antibody system (MP Biomedicals, Solon, Ohio). The estradiol RIA does not cross-react significantly with other estrogens beside estradiol-17β (Estrone: 20% and Estriol 1.51%; all other steroids:<1%) and the intra-assay coefficient of variation is 4.7–10.6%. The testosterone RIA does not significantly cross-react with other androgens beside testosterone (5α-dihydrotestosterone: 3.4%; 5α-androstane-3β, 17β-diol: 2.2%; 11-oxo-testosterone: 2%; all other steroids:<1%) and the intra-assay coefficient of variation is 4.6–9.1%.

## Song recording and analysis

At each MRI imaging time point, songs of all birds were recorded for 2 days, so that we could accurately monitor the evolution of the song behavior in relation to the hormone and neuroplastic changes. Since isolated starlings do not sing as they would do in groups, we temporarily housed the starlings individually in cages that were placed next to the aviary. Each cage was equipped with a Behringer C-2 condensator microphone placed on top, which was connected to a PR8E multi-channel preamplifier (SM Pro audio), 'Delta Series' Breakout box (M-audio) and personal computer with Sound Analysis Pro (SAP) software version 2011.104 (http://soundanalysispro.com/) for automated song detection. Due to technical issues, songs of PR and SD12 of male birds were lost.

For each bird at each time point, song recordings of 4 consecutive hours in the morning were analyzed using Raven Pro 1.5 (Cornell Lab of Ornithology, Ithaca, NY). We created sonograms that were acoustically and visually inspected to identify individual songs (N = 881) from background noise of the aviary based on their intensity and the spectral definition. In line with prior song processing studies (*Eens, 1997*; *Van Hout et al., 2012*), we defined a song bout as a period of at least 5 s of song with pauses no longer than 1.5 s. Starling song consists of a series of distinct song elements or phrases, which can be allocated to four categories of phrase types that are mostly performed in a fixed order. The song often starts with one or several whistles (whistle phrase type), followed by a section of complex phrases (variable phrase type) and rapid series of clicks or rattles (rattle phrase type), before ending the song with high frequency song elements (high-frequency phrases type). Song bouts containing at least three different phrase types were labeled as 'complete song bouts'.

The number of complete song bouts sung per hour within the 4 hr period was taken as a measure of the song rate. We used the first 20 complete song bouts sung within this 4 hr period to calculate the song bout length. This way, we avoided the overrepresentation of songs during time points where the song rate was high.

## MRI data acquisition

The birds were initially anesthetized using 2% Isoflurane (Isoflo , Abbot Laboratories Ltd.) with a mixture of oxygen and nitrogen at flow rates of 100 and 200 cm$^3$/min, respectively. Throughout the imaging procedure, the anesthesia was gradually reduced to 1% Isoflurane in response to their

respiration rate, which was monitored with a small pneumatic sensor (SA Instruments, NY, USA) positioned under the bird. Body temperature was monitored with a cloacal temperature probe and kept within a narrow physiological range ([41.0 ± 0.2] ℃) using a warm air system with a feedback unit (SA Instruments, NY, USA).

All MRI measurements were performed on a 7T horizontal MR system (Pharmascan 70/16 US, Bruker Biospin, Germany). In each imaging session, we acquired a T2-weighted 3D anatomical Rapid acquisition with relaxation enhancement (RARE) scan (3D; TR: 2000 ms; $TE_{eff}$: 44 ms; RARE factor: 8; acquisition matrix of (256 × 92×64) giving a voxel resolution of (0.089 × 0.250×0.250) $mm^3$). Subsequently, a four shot spin-echo echo-planar imaging (SE-EPI) diffusion weighted (DW) scan (TR: 7000 ms; TE: 23 ms; δ 4 ms, Δ 12 ms; b-value 670 $s/mm^2$; 60 diffusion gradient directions; spatial resolution: (0.179 × 0.179×0.350) $mm^3$; 28 coronal slices) was acquired. The 60 diffusion gradient directions were divided over three scans, each starting with the acquisition of 7 b0 images. Total acquisition time of all MRI scans amounted to 50 min. After the scanning session, birds were left to recover in a warmed recovery box before returning to the song recording cages next to the aviary.

## MRI data processing

Whole brain volume was manually delineated on the T2-weighted 3D anatomical RARE scan, which covered the entire brain, including telencephalon, diencephalon, mesencephalon, and metencephalon. These volumes were used as measure of brain size, used in further statistical analysis.

DWI were prepared for voxel-based analysis using MRtrix3 version 3.0 (*Tournier et al., 2012*). We used an in-house algorithm to convert the Bruker 2dseq files to nifti files, which are compatible with other software programs such as SPM and MRtrix3. This step includes a signal scale correction, since the DWI data is acquired in three separate sequential scans. Furthermore, all DW-images were scaled by factor a 10 in three dimensions, to enable proper processing of small brains in software programs designed to process human data like SPM and MRtrix3. Voxel-based analysis requires that all images are spatially normalized to the same template, to enable voxel-wise comparisons. A simplified overview of the different MRI data processing steps is given in *Figure 12A–D*.

### DTI processing

Preprocessing the diffusion data for voxel-based analysis was performed using MRtrix3 (*Tournier et al., 2012*). First, diffusion gradient orientations were checked and automatically corrected to match the coordinate frame of MRtrix3 and ensure the best global 'connectivity' (*Jeurissen et al., 2014a*). Since the DTI data is acquired in three separate sequential scans, we further corrected for intensity differences between scans by rescaling the diffusion scans based on their b0 images. Preprocessing of the individual DW-images included following steps: denoising (*Veraart et al., 2016*), correction for Gibbs ringing (*Kellner et al., 2016*), motion and distortion correction using FSL (*Andersson and Sotiropoulos, 2016*; *Jenkinson et al., 2012*), bias field correction using ANTS (Advanced Normalization Tool; *Avants et al., 2010*), creating automated whole brain mask for whole brain extraction that were manually checked, upsampling to isotropic voxels of 1.75 mm. These preprocessed diffusion weighted images were used to calculate individual diffusion maps (fractional anisotropy, mean, axial and radial diffusivity). The transformation parameters derived from building the FOD template (see 5.5.2) were applied to the diffusion maps to warp them into the template space to perform voxel-based analysis. Next, these images were smoothed to double voxel size (3.5 × 3.5×3.5 $mm^3$). Finally, all normalized diffusion maps were averaged to create a fractional anisotropy template that is used as background to display the statistical results.

### Fixel-based analysis

For calculation of the fiber-based metrics, we followed the preprocessing steps as defined in *Raffelt et al., 2017*. Fixel-based analysis follows the same preprocessing steps as DTI processing up until the bias field correction. Apparent fiber density analysis differs from DTI analysis by the fact that it is related to the diffusion weighted signal intensity within a given voxel. Therefore, global intensity normalization is performed to ensure robust voxel-wise comparison across subjects. Within this step we used the default fractional anisotropy threshold of 0.4 to create an approximate WM mask, which is used to normalize the median white matter b = 0 intensity across all subjects (*Raffelt et al., 2012*). For each image, a white matter response function was estimated for spherical

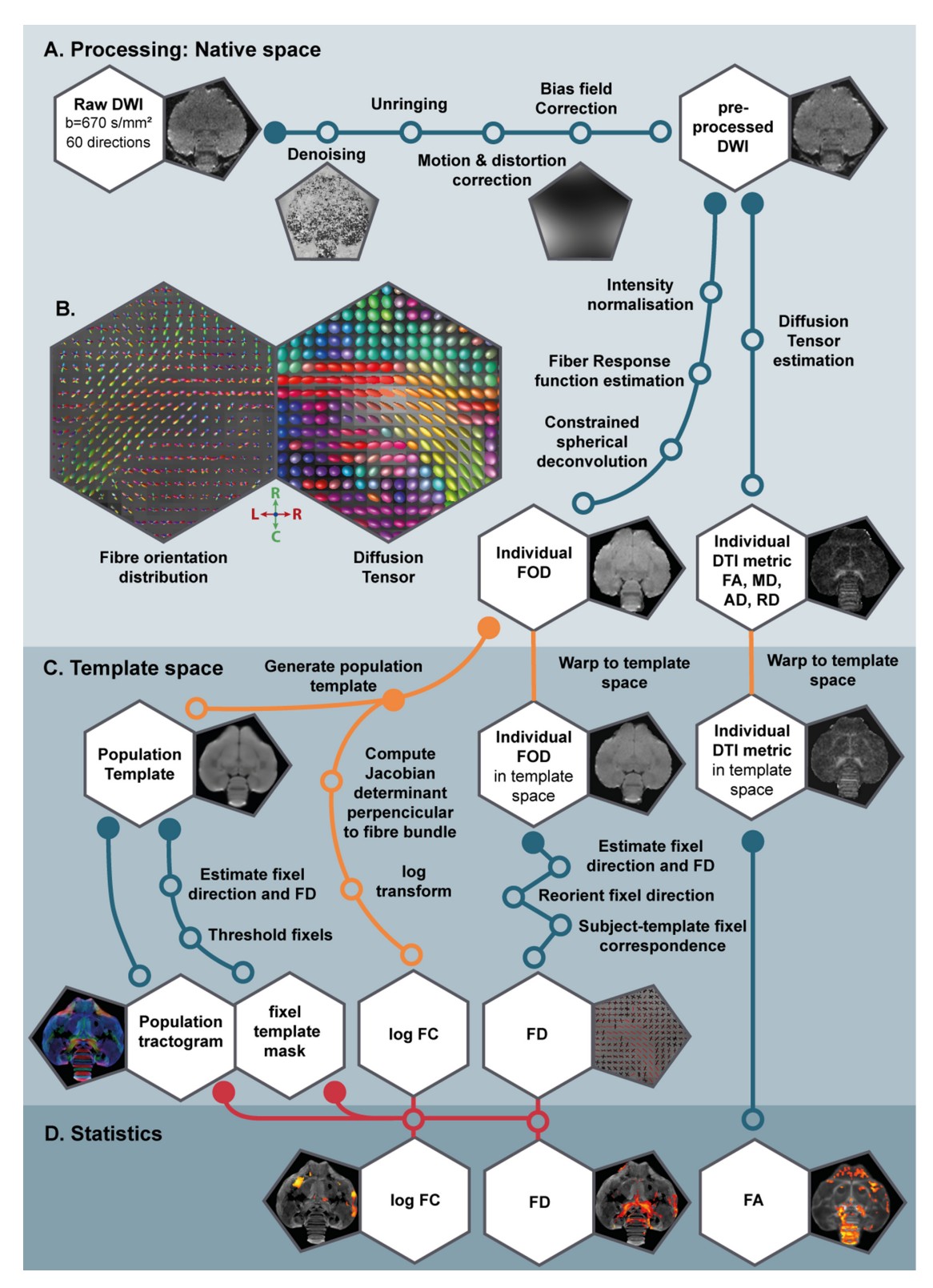

**Figure 12.** Schematic overview of the various processing steps of the DTI and fixel-based analysis. (**A**) Using the DWI, we performed a common DTI analysis and a fixel-based analysis. By doing this, the extracted DTI metrics (such as fractional anisotropy (FA)) and fixel-based metrics (fiber density (FD) and fiber-bundle cross-section (FC)) complemented each other during interpretation of the results. Insets present an example DWI image at each respective processing step. (**B**) Large insets are at the level of the commisura anterior and OM tract and show the fiber orientation distribution (FOD)

*Figure 12 continued on next page*

*Figure 12 continued*

and Diffusion Tensor map used in fixel-based and DTI-based analysis respectively as described in *Table 1* collored according to the standard red-green-blue code (red = left right orientation (L–R), blue = dorso ventral (D–V) and green = rostro caudal (R–C)). (C) All measures were normalized to a population template, a population-based tractogram and fixel template mask were created which are necessary for the voxel- and fixel-based statistical analysis (D). The material and method section contains the extended explanation on the various processing steps. LD, long days; PR, photorefractory state; SD short days.

deconvolution using the unsupervised Dhollander algorithm (*Dhollander et al., 2019*). Next, the average of all individual response functions was calculated and used for constrained spherical deconvolution to estimate FOD images (*Jeurissen et al., 2014b*). These FOD images were normalized to create a unbiased study-based FOD template, which involves linear and non-linear registration (*Raffelt et al., 2011*).

Next, the fixels in the FOD template are thresholded at 0.15, identifying the template white matter fixels to be included in further analysis. This threshold is lower than the default threshold of 0.25 for human brain, as this threshold is too high for the songbird brain and excludes many of the genuine white matter fibers. Unlike the mammalian brain which has a laminar organization with large white matter tracts, the avian cerebrum has a nuclear organization with distinct white matter tracts organized in lamina (*Avian Brain Nomenclature Consortium et al., 2005*). Certain connections between avian nuclei cross the gray matter like pallium and organize in bundles such as the HVC-RA tract (*Karten et al., 2013*). We are aware that choosing a lower threshold comes with the risk of introducing noisy fixels, especially within gray matter, which is a downside of the methodology.

The estimated transformation parameters or warps of each subject to the template were used to transform the individual FOD maps into template space without FOD reorientation, so that the apparent fiber density can be estimated prior to reorienting the fixels. In the next step, we compare the fixels within a spatially matching voxel of the individual subject and the template white matter fixels, to identify which fixels correspond to each other and subsequently assign the corresponding fiber density value (*Raffelt et al., 2017*).

Next to FD, we also computed a fixel-based metric related to the macroscopic morphology in fiber bundle cross-section (FC). Fiber bundle cross-section (FC) information relies solely on the transformation parameters or Jacobian determinants generated during the construction of the population template, similar to other morphometry analyses like voxel-based morphometry (*Ashburner and Friston, 2000*). Morphological differences in the plane perpendicular to the fixel orientation could reflect differences in the number of axons, myelination or the spacing between axons. For group statistical analysis, the FC values were logarithmically transformed to fiber-bundle cross-section to ensure that the data are centered around zero and normally distributed. Positive values indicate then expansion, whereas negative values reflect shrinkage of a fiber bundle relative to the template (*Raffelt et al., 2017*).

## Tractography

A whole-brain tractogram was generated from the FOD template, which is required for statistical analysis using connectivity-based fixel enhancement (CFE) (*Raffelt et al., 2015*). We created 20 million tracks using dynamic seeding, and maximum angle of 45°. We used Spherical-deconvolution informed filtering (SIFT) to filter our tractogram to ensure that the streamline densities approximate the biological fiber densities estimated by spherical deconvolution. SIFT2 creates for each streamline a weighting coefficient based on the underlying FOD (*Smith et al., 2015*). Currently, track-weights created in SIFT2 cannot be implicated in statistical analysis. Therefore, in our statistical analyses, we used SIFT, which selectively filters out streamlines based on the spherical deconvolution to reduce the tractogram to 2 million tracks (*Smith et al., 2013*).

## Statistical analysis

Voxel-based statistical analyses allow for unbiased determination of sexual dimorphisms, longitudinal changes over time and the neural correlates of song behavior (performed using SPM12). To examine group level repeated measurements in a voxel-based manner using the standard general linear model approach as implemented in SPM and FSL, we followed the recommendations of

*McFarquhar, 2019*. Effects of sex and brain size are assessed using full factorial design, whereas the effect of time and the interaction between sex and time are assessed using a flexible factorial design in SPM.

Voxel-based clusters are further explored by extracting the average values from these statistical ROI clusters and investigate either their change over time or their correlation to a certain song parameter. Traditional correlation techniques like Pearson correlation require independence of data. Since we have repeated measures, this assumption of independence is violated when performing a multiple regression on longitudinal data. Traditionally, this is solved by using data from only one time point. However, a lot of information on temporal changes is lost this way. Therefore, we use repeated measures correlation instead of traditional correlation techniques (*Bakdash and Marusich, 2017*). This way, we can assess the within-subject correlation, which gives a meaningful interpretation of how testosterone, song and diffusion parameters correlate to each other on a within-subject level.

Hormone levels, brain size, and song parameters were analyzed using linear mixed model in JMP Pro 15 (SAS Institute Inc, Cary, NC, 1989–2007) to assess the main effect of time, sex and their interaction.

## Song analysis and plasma testosterone and estrogen concentrations

We investigated whether brain size, song behavior, testosterone, and estrogen levels changed significantly over time using linear mixed model ('time point', 'sex' and the interaction 'sex * time point' as fixed factors and 'subject ID' as a random factor) via JMP Pro 15 (SAS Institute Inc, Cary, NC, 1989–2007). Tukey's Honest Significant Difference (HSD) was used for post hoc statistical testing to correct for multiple comparison.

Subsequently, within-subject correlations between testosterone and song parameters were assessed by using the repeated measures correlation package (rmcorr in R) (*Bakdash and Marusich, 2017*). Differences were considered significant for p<0.05.

## Statistical analysis brain size

To investigate the role of brain size on the general sexual dimorphism and on song behavior, we artificially divided the starlings by their brain size using a median split analogous to the analysis in *Kurth et al., 2018*. Birds with a brain size larger than the median brain size of 1805 mm$^3$ were considered as large brain starlings. This way, we divided females in large and small brain groups (respectively, females: N = 6 large and N = 6 small, males N = 6 large, N = 7 small). In order to test whether song or testosterone change differently in large versus small brain birds, we performed a linear mixed model analysis for each sex separately with time, brain size (categorical factor of 2 levels) and their interaction as fixed factors and subject as a random factor. Post hoc testing in the linear mixed model analysis was corrected with HSD for multiple comparison using JMP Pro 15. Brain size (continuous variable), body weight, and song parameters were averaged over time to perform a Pearson correlation between these factors for each sex separately in JMP Pro 15.

## Voxel-based analyses

To identify the general sexual dimorphisms between male and female starlings, we first performed a full factorial voxel-based analysis within the entire brain using SPM12. The same full factorial design was used in the voxel-based analysis for categorical brain size difference with the caveat that brain size was substituted as a factor instead of sex. In a separate analysis, we determined the longitudinal neuroplastic changes over time within the song control system, auditory system and their interconnecting tracts using a flexible factorial analysis. To investigate where within the starling brain microstructural properties relate to song behavior parameters, we executed voxel-based multiple regression analyses for fractional anisotropy. To this end, the averaged song behavior parameters were used for this analysis. Clusters were only considered significant if their peak value reached the significance level of p<0.05 corrected for the familywise error rate (FWE) and contained at least (k$_E$) 10 (contiguous) voxels. Flexible factorial and multiple regression analyses were performed with a small volume mask including the song control nuclei, auditory system and their interconnecting tracts. This way, we ensured we did not miss significant changes in relevant regions that might be obscured by the stringent correction for multiple comparisons in whole brain analysis. Results shown

are from the voxel-based analysis with the latter small volume correction. Results were displayed with a significance threshold of p<0.001 (uncorrected for multiple comparison) and overlaid to a group-average fractional anisotropy map (average of all spatially normalized fractional anisotropy maps, including all subjects and time points).

Significant clusters were converted to ROIs (cluster-based ROIs) of which the average diffusion parameters (fractional anisotropy, mean, axial and radial diffusivity) and fixel-based parameters (fiber density and fiber-bundle cross-section) were extracted and plotted using GraphPad Prism. In order to determine the extent of the overall association between diffusion parameters and song rate, Pearson's correlation was calculated using JMP Pro 15 (SAS Institute Inc, Cary, NC, 1989–2007). Further in depth analysis was performed to determine if this overall association was also caused by a within-subject correlation by using rmcorr in R (*Bakdash and Marusich, 2017*).

### Fixel-based analysis

Fixel-based statistical analysis was performed using the connectivity-based fixel enhancement (CFE) described by *Raffelt et al., 2015*. A connectivity matrix was calculated using the fixel template and SIFT filtered tractogram. This connectivity matrix was used to perform connectivity-based smoothing where only fixels belonging to the same fiber tract are smoothed, instead of the neighboring voxels as in a voxel-based analysis. A fixel-based analysis was performed to assess the difference between sexes and brain size, both examined in separate tests. In the CFE statistical test, we used whole block exchangeability to look for changes between subjects and included the time point information in a variance group. Similarly to the voxel-based DTI analysis, statistical results were visualized with a significance threshold of p<0.001 (uncorrected for multiple comparison) and color-coded according to their significance T-value or the physical direction of the tract.

## Acknowledgements

The authors wish to thank Ben Jeurissen (Vision Lab of the University of Antwerp) and Robert Smith (The Florey Institute of Neuroscience and Mental Health) for their help with MRtrix3. This research was funded by a grant from the Research Foundation – Flanders (FWO, project Nr G030213N) and Interuniversity Attraction Poles (IAP) ('PLASTOSCINE': P7/17) to Annemie Van der Linden. Jasmien Orije is a PhD fellow of the FWO (Nr 1115217N) and Elisabeth Jonckers received Postdoctoral fellowships of the FWO (Nr 12 R1917N).

## Additional information

### Funding

| Funder | Grant reference number | Author |
|---|---|---|
| Belgian Federal Science Policy Office | "PLASTOSCINE": P7/17 | Annemie Van der Linden |
| Research Foundation Flanders | G030213N | Annemie Van der Linden |
| Research Foundation Flanders | 115217N | Jasmien Orije |
| Research Foundation Flanders | 12R1917N | Elisabeth Jonckers |

The funders had no role in study design, data collection and interpretation, or the decision to submit the work for publication.

### Author contributions

Jasmien Orije, Data curation, Formal analysis, Investigation, Visualization, Writing - original draft, Writing - review and editing; Emilie Cardon, Data curation, Formal analysis, Writing - review and editing; Julie Hamaide, Investigation, Methodology, Writing - review and editing; Elisabeth Jonckers, Supervision, Writing - review and editing; Veerle M Darras, Formal analysis, Supervision, Investigation, Writing - review and editing; Marleen Verhoye, Conceptualization, Supervision, Methodology, Writing - review and editing; Annemie Van der Linden, Conceptualization, Resources, Supervision,

Funding acquisition, Investigation, Writing - original draft, Project administration, Writing - review and editing

## Author ORCIDs
Jasmien Orije (iD) https://orcid.org/0000-0002-6699-6221
Veerle M Darras (iD) http://orcid.org/0000-0003-4429-8868
Annemie Van der Linden (iD) http://orcid.org/0000-0003-2941-6520

## Ethics
Animal experimentation: The housing and experimental procedures were performed in agreement with the European directives, Belgian and Flemish laws and were approved by the Committee on Animal Care and Use of the University of Antwerp, Belgium (2014-52).

## Decision letter and Author response
Decision letter https://doi.org/10.7554/eLife.66777.sa1
Author response https://doi.org/10.7554/eLife.66777.sa2

# Additional files
## Supplementary files
• Transparent reporting form

## Data availability
All MRI data have been deposited in Dryad (https://doi.org/10.5061/dryad.h44j0zpj8). Source data files have been provided for Figures 3-11.

The following dataset was generated:

| Author(s) | Year | Dataset title | Dataset URL | Database and Identifier |
|---|---|---|---|---|
| Orije J, Cardon E, Hamaide J, Jonckers E, Darras V, Verhoye M, Linden A | 2021 | The photosensitive phase acts as a sensitive window for seasonal multisensory neuroplasticity in male and female starlings | https://doi.org/10.5061/dryad.h44j0zpj8 | Dryad Digital Repository, 10.5061/dryad.h44j0zpj8 |

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
