## [Decision Letter]

**Acceptance summary:**

The strengths of the study are summarised in the previous decision letter from which it will be clear the referees were very impressed with the study but required revisions for clarity of exposition for this broad audience. The suggestions have been tackled with exemplary thoroughness.

**Decision letter after peer review:**

Thank you for submitting your article "The photosensitive phase acts as a sensitive window for seasonal multisensory plasticity in male and female starlings" for consideration by *eLife*. Your article has been reviewed by 3 peer reviewers, and the evaluation has been overseen by a Reviewing Editor and Timothy Behrens as the Senior Editor. The following individuals involved in review of your submission have agreed to reveal their identity: Tyler Stevenson (Reviewer #1); Alessandro Gozzi (Reviewer #2).

The reviewers were very impressed with this study. They have discussed their reviews with one another, and the Reviewing Editor has drafted this to help you prepare a revised submission. These are detailed as possible revisions below for you to consider

Possible revisions (for the authors):

1. Referee #1 suggest a reconsideration of the title.

2. Please state if any other measures of song complexity were used.

3. Age and sex of birds needs clarification.

4. Referee #2 makes a number of suggestions regarding technical exposition in their public review.

5. Referee #3 raises some question about how the data relate to the known song system in European starlings and suggests an additional figure.

*Reviewer #1 (Recommendations for the authors):*

1. The Raven Pro software is a powerful tool to analyse the complexities of song. The main measures used to evaluate song are rate and bout length. The song rate is primarily driven by hypothalamic structures not highlighted in the manuscript. Bout length is a good measure of song quality, but not complexity. The study would be strengthened if the authors included other measures (e.g. entropy) and then relate any variation to known brain structures (e.g. HVC, Area X).

2. The sex and age of the birds should be clarified. There are a couple statements on this issue in the manuscript (i.e. lines 615-617 and lines 743-745). The interpretation is also slightly limited as the location and time of year for bird capture is not clear. We both males and female collected in France and Cyprus, or were males collected in France and females in Cyprus? The 'respectively' on line 798 does not clearly resolve the confusion. Based on later text, it seems that males were collected in 2014 and females in 2017. The difference in female brain size and body mass is likely due to age and the authors might want to compare their data with Sockman et al., 2004 Biol Reprod. 71:979-986. Also, how might age-effect impact the sex differences observed in the DTI analyses?

3. The Tables include significance values as 0.000. This is not acceptable as the study examines biological variation that overlaps. If the p-value is very low, then simply change to p<0.001 or similar.

*Reviewer #3 (Recommendations for the authors):*

1. Some hot areas in the figures are not labeled, what is the reason for it?

2. It might be useful to show an example of a statistically different area (measured by one or more methods) and then the table describing all the regions that were found with their statistics. Then some of the main figures could go to the supplementary information and the paper itself will be clearer.

3. It shows that the authors put a great amount of effort in the study, and the analysis is very impressive. I believe that the paper in this form may be a little bit too long and it misses the emphasis of novel results (like the involvement of the hippocampus in singing female starling), which could set as ground for further studies. A better introduction of the brain region that will later be described as significant together with some modification to the figures, will make the paper much more appealing to the broader neuroscience community, in my opinion.

---

## [Author Response]

Reviewer #1 (Recommendations for the authors):1. The Raven Pro software is a powerful tool to analyse the complexities of song. The main measures used to evaluate song are rate and bout length. The song rate is primarily driven by hypothalamic structures not highlighted in the manuscript. Bout length is a good measure of song quality, but not complexity. The study would be strengthened if the authors included other measures (e.g. entropy) and then relate any variation to known brain structures (e.g. HVC, Area X).

This study indeed also analyzed more than just song bout length. We included the different phrase types (wistles, variables, rattles and high frequency phrase) used within each song, and a measure for phrase diversity within a song (the different phrase types used within a song bout). Unfortunately, we could not analyze entropy, since song recordings had to be done next to the large aviary (in order for the birds to actually sing) which results in background noise that compromises the entropy calculations.

The finding that the brain size difference was confounded by the difference in singing behavior of female starlings was important for the interpretation of the brain size and sex differences. Which is why we included the song analysis and the neural correlates in this paper. Initially we added the complete and extended song analysis dataset in the paper but it further elongated this already extensive paper, without contributing to the main message. That’s why we opted to have the extended song analysis on song complexity in a separate paper, since the original paper was already becoming quite long and the extra analysis could distract further from the main message. The further extended analysis of female song compared to male song deserves a separate paper on its own, as these findings are relevant for the songbird research community.

2. The sex and age of the birds should be clarified. There are a couple statements on this issue in the manuscript (i.e. lines 615-617 and lines 743-745). The interpretation is also slightly limited as the location and time of year for bird capture is not clear. We both males and female collected in France and Cyprus, or were males collected in France and females in Cyprus? The 'respectively' on line 798 does not clearly resolve the confusion. Based on later text, it seems that males were collected in 2014 and females in 2017. The difference in female brain size and body mass is likely due to age and the authors might want to compare their data with Sockman et al., 2004 Biol Reprod. 71:979-986. Also, how might age-effect impact the sex differences observed in the DTI analyses?

Indeed Males were collected in France in November 2014 and females were collected in Cyprus in 2017. Since both sexes were caught as adults, we do not have their exact age. None of them had brown plumage so they all were at least one year old. We want to clarify that even though both sexes were caught at different times, they all went through the same artificial photoperiodic and MRI acquisitions occurred within a year after catching them. Any age differences that are present within the groups, are out of our control, as we do not know the age of the birds upon catching them in the wild.

We do believe that age is a contributing factor to the difference in brain size as explained in the discussion.

“Brain size is affected by multiple factors including age, circulating hormone levels, environment and experience (Healy and Rowe 2007). Of note, because we used wild caught starlings we cannot control for their exact age or developmental history. The larger females could be older compared to the smaller female starlings.”

We clarified the origin of the animals as follows in line 816:

“All experiments were performed on thirteen male and twelve female starlings (*Sturnus vulgaris*). The males were wild caught as adults in Normandy (France) in November 2014 and the females were caught in Nicosia (Cyprus) in March 2017.”

3. The Tables include significance values as 0.000. This is not acceptable as the study examines biological variation that overlaps. If the p-value is very low, then simply change to p<0.001 or similar.

All tables have been adjusted accordingly.

Reviewer #3 (Recommendations for the authors):1. Some hot areas in the figures are not labeled, what is the reason for it?

We tried to label as many areas as possible, without making the figures too loaded with information. We added some extra labels in figures 3-6, to further clarify some areas. However, labeling is sometimes difficult, since significant clusters blend from one region to the next. For this reason we included the population-based template with the anatomical location of the most important structures in the original manuscript. This can be used as a look up figure to situate the location of a cluster relative to known regions.

2. It might be useful to show an example of a statistically different area (measured by one or more methods) and then the table describing all the regions that were found with their statistics. Then some of the main figures could go to the supplementary information and the paper itself will be clearer.

We asked some further explanation about this recommendation:

To which figures does this statement apply?

Does the reviewer suggest we only select 1 area (with a sex difference, brain size difference, time difference or correlation) to visualize and have the rest of the statistics in table format and supplementary figures?

And if so it might be difficult to choose which area to show, since there are many interesting regions. Furthermore, one of the points we make is that MRI has a whole-brain approach, and we do not want to limit ourselves to for example only the song control system as so many papers before did.

We could for example select HVC, and show all the statistics for that region, but this would maybe bias the message of the paper. And while most areas follow a similar change over time in FA for example, there are subtle differences in when the first changes are notable, and what are the contributing factors.

We understand that there is a balance too be made, so that the paper does not become overwhelming, but this is a difficult one.

This was the reviewer’s response: I accept the authors' answer to my comment, and I think it is reasonable to keep the figures as is. Since I am not an expert in fMRI, I was interested in knowing how the raw data looks like (from which you derive your metrics). If that is something you could easily add as a supplemental figure, it would be great.

An example of the raw images is depicted in figure 12, where we show how an image is processed at each different step. The extracted metrics are now supplied in supplementary figures to figure 3-6 as suggested by reviewer 2.

3. It shows that the authors put a great amount of effort in the study, and the analysis is very impressive. I believe that the paper in this form may be a little bit too long and it misses the emphasis of novel results (like the involvement of the hippocampus in singing female starling), which could set as ground for further studies. A better introduction of the brain region that will later be described as significant together with some modification to the figures, will make the paper much more appealing to the broader neuroscience community, in my opinion.

We now added a few lines to the introduction to introduce the role of other regions like the cerebellum and hippocampus.

However, in vivo Magnetic Resonance Imaging (MRI) allows us to visualize the entire brain and study the involvement of other structures like the hippocampus and cerebellum, which recently have been shown to be involved in song preference and song processing, respectively (Person, Gale et al. 2008, Bailey, Wade et al. 2009, Striedter 2016, Pidoux, Le Blanc et al. 2018).

In addition we added a schematic overview in figure 1 that visualized the connections of the cerebellum and hippocampus together with the scheme of the auditory and song control system.